# Impacts of large-scale atmospheric circulation changes in winter on Black Carbon transport and deposition to the Arctic

Luca Pozzoli[1], Srdan Dobricic[1], Simone Russo[2], and Elisabetta Vignati[1]

[1]European Commission, Joint Research Centre (JRC), Directorate for Energy, Transport and Climate, Air and Climate Unit, Ispra (VA), 21027, Italy
[2]European Commission, Joint Research Centre (JRC), Directorate for Competences, Modelling, Indicators and Impact Evaluation Unit, Ispra (VA), 21027, Italy

*Correspondence to*: Luca Pozzoli (luca.pozzoli@ec.europa.eu)

**Abstract.** Winter warming and sea ice retreat observed in the Arctic in the last decades may be related to changes of large scale atmospheric circulation pattern, which may impact the transport of black carbon (BC) to the Arctic and its deposition on the sea ice, with possible feedbacks on the regional and global climate forcing. In this study we developed and applied a statistical algorithm, based on the Maximum Likelihood Estimate approach, to determine how the changes of three large scale weather patterns associated with winter increasing temperatures and sea ice retreat in the Arctic, impact the transport of BC to the Arctic and its deposition. We found that two atmospheric patterns together determine a decreasing winter deposition trend of BC between 1980 and 2015 in the Eastern Arctic while they increase BC deposition in the Western Arctic. The increasing BC trend is mainly due to a pattern characterized by high pressure anomaly near Scandinavia favouring the transport in the lower troposphere of BC from Europe and North Atlantic directly into to the Arctic. Another pattern with high pressure anomaly over the Arctic and low over the North Atlantic Ocean has a smaller impact on BC deposition, but determines an increasing BC atmospheric load over the entire Arctic Ocean with increasing BC concentrations in the upper troposphere. The results show that changes in atmospheric circulation due to polar atmospheric warming and reduced winter sea ice significantly impacted BC transport and deposition. The anthropogenic emission reductions applied in the last decades were, therefore, crucial to counterbalance the most likely trend of increasing BC pollution in the Arctic.

## 1 Introduction

The Arctic has warmed during the recent decades more strongly than other regions due to the polar amplification of the global warming signal (Serreze and Barry, 2011). Satellite observations show a decline in summer sea ice extent since the late 1970s with a dramatic acceleration in the last decade. The sea ice is also characterized by increased fraction of thinner and younger sea ice, which is melting during the polar summer (Comiso, 2012; Kwok and Rothrock, 2009; Kwok, 2009; Maslanik et al., 2011; Rothrock et al., 2008). The rate of change of the Arctic sea ice has been so fast in the last decades, that the expressions "new Arctic" recently started to be used (e.g. Doscher et al., 2014). Also during winter the sea ice cover and

concentration has changed significantly: a decreasing trend of sea ice concentration in winter months, December-January-February (DJF), for the period 1979-2014, was observed by satellites over the Barents Sea up to a rate of 20% per decade. Several studies investigated whether the varying winter sea ice cover over the Barents Sea may further produce hemispheric scale impacts in the atmosphere with changes in the large scale atmospheric circulation (e.g. Barnes et al., 2014; Cohen et al., 2014; Deser et al., 2007; Mori et al., 2014; Overland and Wang, 2010; Petoukhov and Semenov, 2010; Screen et al., 2013; Semenov and Latif, 2015). Dobricic et al. (2016) analysed the atmospheric reanalysis of the past decades with a statistical approach, the Independent Component Analysis (ICA), which has not been applied frequently in climate studies. ICA can be an efficient methodology to extract independent components, i.e. sharing the minimum information without the gaussianity assumption (Hyvärinen and Oja, 2000). Dobricic et al. (2016) found that three independent atmospheric patterns, connected to the North Atlantic Oscillation (NAO), the Scandinavian Blocking (SB), and the El Nino Southern Oscillation (ENSO), are closely related to the ongoing hemispheric increase of near-surface temperature over the Barents Sea and may be related to the sea ice cover shrinkage in this region. Thus, the changes in the ocean-sea ice-atmosphere energy fluxes may also impact the large scale atmospheric circulation in the North Hemisphere, as well as the transport of atmospheric pollutants with possible feedbacks on the regional and global climate.

The aim of this study is to estimate how these changes in the large scale circulation patterns of the North Hemisphere in winter could affect the transport of pollutants to the Arctic, and in particular black carbon (BC). BC is a short lived climate pollutant (SLCP) and thus one of the key targets for emission reduction strategies to mitigate the effects of climate change (Shindell et al., 2012). BC plays a key role in the Arctic climate system as it affects the Earth radiative balance through different mechanisms. BC particles directly absorb solar radiation, and above a surface with high albedo, such as snow and sea ice, they warm the atmosphere inside the haze layer and at higher altitudes (Shaw and Stamnes, 1980). On the other hand, BC may also cool the Arctic climate due to surface dimming and decreased poleward heat flux caused by weakened latitudinal temperature gradient from BC heating of the upper troposphere (Flanner, 2013; Sand et al., 2013a, 2013b; Shindell and Faluvegi, 2009). An indirect BC radiative forcing process is determined by its deposition on snow and ice, which can accelerate their melting. Hadley and Kirchstetter (2012) measured in laboratory how BC snow contamination can reduce snow albedo, with amplified BC radiative perturbation in larger snow grains size, which is also consistent with the parameterizations of BC and soot concentrations in snow included in climate models (Aoki et al., 2000; Flanner and Zender, 2006; Yasunari et al., 2011). Due to lower insolation in the Arctic during winter and early spring, BC exerts a negligible radiative forcing, but particles that deposit to snow and ice surfaces can re-emerge at the surface when melt commences in the summer (Conway et al., 1996), meaning that winter transport and deposition of BC is also affecting the Arctic climate.

Measurements show that the equivalent BC (EBC, filter based absorption measurements of aerosol particles) surface concentrations in the Arctic, as well as those of other atmospheric pollutants, such as sulphate, are largest in winter and early spring, when the transport of pollutants from lower latitudes is more efficient and the removal processes slower (e.g. Eleftheriadis et al., 2009; Gong et al., 2010; Hirdman et al., 2010; Sharma et al., 2006, 2013). The extreme cold surface temperatures in the Arctic determine strong inversions building the so-called polar dome (Klonecki, 2003; Stohl, 2006),

characterized by stable air masses inside the dome and limited exchange between the boundary layer and the free troposphere, meaning that in the Arctic local sources of pollution are transported more efficiently. On the other hand the long range transport of BC from lower latitudes may contribute more to the Arctic Haze due to the significantly larger mid-latitudes emission levels compared to those in the Arctic region (Sand et al., 2015). Another barrier which isolate the Arctic from the lower latitudes is the Arctic front (Barrie, 1986). Polluted air masses can reach the Arctic lower troposphere only if they are emitted in a sufficiently cold region located north of Arctic front, a situation which mainly occurs during winter in north Eurasia. Pollution emitted south of the Arctic front is lifted above it, generally with cloud formation and scavenging of aerosol particles and deposition to the surface by precipitation. Pollutants emitted in North America and East Asia can reach the Arctic mainly through this pathway (Stohl, 2006). As a result the main contribution to winter BC surface concentrations and deposition in the Arctic are originated from northern Eurasia (Hirdman et al., 2010). The BC concentration and deposition vary considerably also on inter-annual time scale, and large scale atmospheric circulation processes can favour or reduce the transport of pollutants from the main source regions (North America and Eurasia) toward the Arctic. Hirdman et al. (2010) analysed the long-term trends of EBC and sulphate measured at three stations in the Arctic. They found that EBC surface concentrations decreased at two Arctic stations, Alert (Canada) and Zeppelin (Svalbard islands, Norway), while there is no trend detectable at Barrow (Alaska). Analysing the atmospheric back trajectories at the three stations, they concluded that the observed decreasing trends are mainly driven by changes in the emissions, while the impact of atmospheric circulation can only explain a minor fraction of the downward trend. They found significant correlations between the North Atlantic Oscillation Index (NAOI) and air masses from North America and North Eurasia at both Alert and Barrow, but not for the station at Zeppelin. However, the full understanding of how the transport changes related to large scale circulation patterns, such as the NAO, impact the BC concentrations and deposition to the Arctic has not yet been established.

In this study we apply a statistical methodology, based on a Bayesian approach, to investigate the relationships between large scale atmospheric circulation patterns and the long range atmospheric transport of air pollutants to the Arctic in recent decades. We estimate the most likelihood BC distribution associated with large scale atmospheric patterns which can approximate the near-surface temperature trend in the Arctic, both spatially and temporally, of two atmospheric reanalysis of the past decades (Dobricic et al., 2016). The distributions of BC surface concentration, total column and deposition fluxes were taken from a hindcast simulation of tropospheric chemistry composition for the period 1980-2005. Three different BC simulations are analysed, one with changing anthropogenic emissions for the entire period, and two with fixed anthropogenic emissions of the year 1980 and 2000, respectively. With this methodology we aim to quantify the trends and inter-annual variability of BC surface concentrations, load and deposition in the Arctic associated with winter large scale atmospheric circulation changes occurred in the last decades.

## 2 Methods and data

In the following sections we will first describe the statistical method (Section 2.1) and the data used for the analysis. Section 2.2 provides a short introduction of the large scale atmospheric patterns identified by (Dobricic et al., 2016), which are mainly contributed to the winter polar warming of the last decades. Section 2.3 describes the data used to estimate the maximum likelihood distribution of BC concentration and deposition associated with the specific atmospheric patterns.

### 2.1 Maximum Likelihood Estimate of atmospheric pollutants

The Maximum Likelihood Estimate (MLE) of the distribution of a pollutant in the atmosphere associated with a specific atmospheric pattern may be derived starting from the Bayes theorem. The probability, *p(c/a)*, of an unknown joint distribution of an atmospheric pollutant and corresponding atmospheric state, **c**, given a specific atmospheric pattern, **a** , may be expressed as:

$$p(\mathbf{c}\,|\,\mathbf{a}) \cong p(\mathbf{a}\,|\,\mathbf{c})p(\mathbf{c}). \tag{1}$$

where *p(c)* is the priori probability of coupled pollutant distribution (e.g. of atmospheric concentrations, surface deposition or atmospheric burden) coupled with corresponding atmospheric conditions, and *p(a/c)* is the probability that a certain atmospheric pattern (e.g. of surface pressure, geopotential height, wind speed) is associated with the atmospheric state coupled with pollutant distributions. The prior probability, *p(c),* can be estimated for example by using chemistry-climate model simulations of the past or future atmospheric physical parameters and the corresponding coupled chemical composition**s**. In order to simplify the mathematical solution of the problem we assume that the two distributions, *p(a/c)* and *p(c),* are approximately Gaussian, meaning that also their product, *p(c/a)*, will be Gaussian. This assumption is supported by the Central Limit Theorem, which tells that the distribution produced by several processes with non-Guassian distributions should appear closer to a Gaussian distribution (e.g. Hyvärinen and Oja 2000). Thus it is possible to estimate the probability of a pollutant distribution for any atmospheric pattern **a$_k$**, which can be taken for example from an independent atmospheric model simulation, or a specific large scale atmospheric pattern, like the ENSO or the NAO. Equation (1) becomes:

$$p(\mathbf{c}\,|\,\mathbf{a}_k) = const \times \exp\left\{-\frac{1}{2}[\mathbf{a}_k - H(\mathbf{c})]^T \mathbf{D}^{-1}[\mathbf{a}_k - H(\mathbf{c})] - \frac{1}{2}\mathbf{c}^T \mathbf{C}^{-1}\mathbf{c}\right\}, \tag{2}$$

where, **H(c)** is a mapping function between coupled atmospheric states being a part of **c** and the specific atmospheric pattern $\mathbf{a}_k$ that may be defined over different model grids, **D** is the covariance matrix of the differences between atmospheric anomalies, while **C** is the covariance matrix of the coupled atmosphere and pollutant anomalies.

The Maximum Likelihood Estimate (MLE) of $p(\mathbf{c}\,|\,\mathbf{a}_k)$ is the one with the minimum absolute value of the argument of the exponential function in Equation (2) that is the minimum of the cost function *J*:

$$J = \frac{1}{2}\left[\mathbf{a}_k - H(\mathbf{c})\right]^T \mathbf{D}^{-1}\left[\mathbf{a}_k - H(\mathbf{c})\right] + \frac{1}{2}\mathbf{c}^T\mathbf{C}^{-1}\mathbf{c}. \tag{3}$$

Assuming that the mapping $H$ is linear (e.g. a bilinear interpolation, $\mathbf{H(c)} = \mathbf{Hc}$), $J$ becomes a quadratic function with a single minimum which may be estimated by a linear minimizer. The numerical stability may be further increased by defining a control subspace $\mathbf{z} = \mathbf{Z}^+\mathbf{c}$, where $\mathbf{Z}$ is a square root of $\mathbf{C}$ and the superscript + indicates the generalized inverse. The cost function becomes:

$$J = \frac{1}{2}\left[\mathbf{HZz} - \mathbf{a}_k\right]^T \mathbf{D}^{-1}\left[\mathbf{HZz} - \mathbf{a}_k\right] + \frac{1}{2}\mathbf{z}^T\mathbf{z} \tag{4}$$

After finding $\mathbf{z}$, the most likelihood coupled anomaly is obtained from:

$$\mathbf{c} = \mathbf{Zz}. \tag{5}$$

The cost function $J$ is minimized using the quasi-Newton Limited memory Broyden–Fletcher–Goldfarb–Shanno (L-BFGS) minimizer (Byrd et al., 1995). It is assumed that $\mathbf{D}$ is a diagonal matrix and elements along the diagonal have equal variance divided by the area of each grid point. The atmospheric patterns $\mathbf{a_k}$, for which we estimate the maximum likelihood distribution of atmospheric pollutants, are those previously described by Dobricic et al. (2016) and briefly described in Section 2.2. Matrix $\mathbf{C}$ is estimated from coupled anomalies of a multi-year simulation with a coupled atmosphere-chemistry model, described in more details in Section 2.3. Coupled anomalies contain both the atmospheric physical parameters and pollution concentrations. In order to filter out statistically insignificant relationships, the covariance matrix of anomalies is approximated by forming the Empirical Orthogonal Function (EOF) decomposition and by maintaining only EOFs with major eigenvalues. The minimum of the cost function $J$ is found for each winter month (from December to February, DJF) and for each atmospheric pattern $\mathbf{a_k}$, separately.

## 2.2 Atmospheric patterns

Dobricic et al. (2016) performed an Independent Component Analysis (ICA) of atmospheric reanalysis data finding a link between the increasing trend of near surface temperature in the Arctic during winter (December, January, and February, DJF) and three atmospheric patterns. Hannachi et al. (2009) first proposed to apply the ICA instead of the commonly used EOF in climate studies showing that ICA may be explained by a rotation of EOFs. The major difference between the two estimates is that ICA does not assume the Gaussian distributions of event probabilities. This property ensures that components are truly independent and not just uncorrelated, on the other hand it is not possible to determine the order of the independent components. In Dobricic et al. (2016) the Fastica algorithm by Hyvärinen and Oja (2000) was applied to extract independent atmospheric components in winter during the period 1980-2015. The ICA algorithm finds the best approximation of a matrix $\mathbf{X}$ containing rows of temporal anomalies in the physical space, as:

$$X \cong AS \tag{6}$$

where the columns of matrix **A** represent spatially varying intensities and the rows of the orthogonal matrix **S** are the temporally varying independent components. Matrix A is full rank, but its columns are not orthogonal. This means that there may be overlapping spatial features in different columns of A. Details on implementing ICA for large-scale atmospheric processes may be found in Dobricic et al. (2016), here we introduce the main results of their study, which are used as an input for our analysis. A set of three independent large scale atmospheric structures were found, with significant linear trends and together approximating the spatial variability of near surface temperature trend during winter, as well as atmospheric anomalies of wind intensity, temperature and geopotential height at different pressure levels, from surface up to 10 mbar. Figure 1 shows the spatial distribution of the trends from 1980 to 2015 averaged over the three winter months for near surface temperature at 1000 mbar (T1000) and geopotential height at 850 mbar (H850). The spatial distribution of the three independent components (ICs) with statistically significant trends were named by visually recognizing their similarity to well-known large-scale weather patterns, the North Atlantic Oscillation (NAO), Scandinavian Blocking (SB), and the El Nino-Southern Oscillation (ENSO). In this manuscript we will refer to the IC patterns as $IC_{NAO}$, $IC_{SB}$, and $IC_{ENSO}$ to avoid confusion with the NAO, SB and ENSO indices. The mean winter H850 trend of $IC_{NAO}$ (Figure 1a) with increasing geopotential height over the Arctic and decreasing in the Atlantic Ocean near the Azores, clearly appears as a tendency toward the negative phase of the NAO. T1000 increases over the Arctic with maximum over western Greenland, the Canadian archipelago, and the Barents Sea, at the same time it decreases over northern Europe and Siberia (Figure 1d). The $IC_{SB}$ trend shows an increasing geopotential height over Scandinavia and northwestern Siberia which indicates a prevailing anticyclonic anomaly over the area, bringing warm air from Europe to the Arctic, and vice versa cold air from the Arctic to Eurasia (T1000 in Figure 1e). As shown in Figure 1c,f $IC_{ENSO}$ has a small T1000 trend over the Arctic, compared to the other two IC patterns. The reanalysis trends of T1000 and H850 are well approximated by summing the three ICs, all dominant features are captured both spatially and temporally, in particular the prominent dipole between the strong warming in the Arctic and cooling over Siberia (Dobricic et al., 2016). Consistent results were obtained using two different atmospheric reanalysis (Dobricic et al. 2016), the National Center for Environmental Prediction (NCEP, Kalnay et al., 1996), and ERA-Interim (Dee et al., 2011) from the European Centre for Medium-Range Weather Forecasts (ECMWF). In our study we will estimate the maximum likelihood distribution of BC atmospheric concentration, load and deposition associated with the $IC_{NAO}$ and $IC_{SB}$ atmospheric patterns found in Dobricic et al. (2016). We do not discuss the third atmospheric pattern, the $IC_{ENSO}$, which showed weaker connection to the changes seen in near-surface temperature and geopotential height in the Arctic region.

**2.3 Atmospheric chemical composition**

We used winter monthly mean anomalies (DJF) of BC concentrations and surface deposition fluxes of the period 1980-2005 from hindcast simulations of the fully coupled aerosol-chemistry-climate model, ECHAM5-HAMMOZ (Pozzoli et al., 2011), which is composed of the general circulation model ECHAM5 (Roeckner et al., 2003), the tropospheric chemistry and aerosol module HAMMOZ (Pozzoli et al., 2008a). The horizontal resolution of the simulations is about 2.8°× 2.8°, with

31 vertical levels from the surface up to 10 hPa. The simulation was forced by nudging the reanalysis meteorological fields from the ECMWF ERA-40 re-analysis (Uppala et al., 2005) from year 1980 until 2000 and the operational analyses (IFS cycle-32r2) was used until year 2005. The dry deposition scheme of aerosol particles follows Ganzeveld and Lelieveld (1995), while in-cloud and below cloud scavenging follows the scheme described by Stier et al. (2005). The anthropogenic

emissions of CO, NOx, and VOCs for the period 1980–2000 are taken from the RETRO inventory (Endresen et al., 2003; Schultz et al., 2007, 2008). The AeroCom hindcast aerosol emission inventory (Diehl et al., 2012) was used for the annual total anthropogenic emissions of primary black carbon (BC), organic carbon (OC) aerosols and sulfur dioxide (SO2). The BC anthropogenic emissions remained almost constant globally during the simulated period (1980-2005), 4.9 Tg/year, however large changes occurred in North America, Europe, Former Soviet Union (FSU) and East Asia (Figure S1a,c). Most

of the anthropogenic BC is emitted between 30°N and 60°N, and decreased after the 1990s from about 3 Tg/year to about 2.6 Tg/year after 2000. Above 60°N BC anthropogenic emissions are a small fraction of the total and decreased from 100 to 30 Gg/year. The simulation includes also inter-annual varying biomass burning emissions, from tropical savannah burning, deforestation fires, and mid-and high latitude forest fires published by Schultz et al. (2008). BC biomass burning emissions are ranging between 10 and 170 Gg/year above 60°N, and between 35 and 460 Gg/year at mid-latitudes, with peak years

connected also to inter-annual meteorological variability (Figure S1b,d).

The model has been extensively evaluated in previous studies by comparing simulated chemical concentrations and physical parameters to observations (Auvray et al., 2007; Pausata et al., 2012, 2013; Pozzoli et al., 2008a, 2008b; Rast et al., 2014; Stier et al., 2005; Bourgeois and Bey, 2011) and within model inter-comparison studies (Kim et al., 2014; Pan et al., 2015; Tsigaridis et al., 2014). As shown by Bourgeois and Bey (2011), ECHAM5-HAMMOZ largely underestimate BC

concentrations over the Arctic, both near the surface as well as in the atmospheric column. Compared to the BC measurements from SP2 instrument (Moteki and Kondo, 2007; Schwarz et al., 2006) of the Arctic Research of the Composition of the Troposphere from Aircraft and Satellites (ARCTAS, Jacob et al., 2010), the simulated BC concentrations show a mean absolute bias in the troposphere of 95% in spring, while in summer the BC is well simulated in the upper troposphere and overestimated by 50% near the surface. Bourgeois and Bey (2011) identified the wet scavenging of aerosol

particles as one of the main processes responsible for model bias in winter, a model simulation with revisited wet scavenging coefficients from Henning et al. (2004), considerably improved the simulated BC concentrations in the troposphere in winter, reducing the mean absolute bias to 38%. The large bias of simulated BC and EBC concentrations in the Artic is a known issue, shared with several global climate and chemical tranport models (Eckhardt et al., 2015; Sand et al., 2017). Qi et al. (2017) estimated that the Wegener-Bergeron-Findeisen (WBF) process in mixed-phase clouds icreases BC in the

atmosphere by 25% to 70% by reducing wet scavenging efficiency. Other factors which may improve the simulated BC distirbution in the Artcic are dry deposition velocities calculated with resistence-in-series method over all surfaces (ocean, snow/ice) and improved BC flaring emissions (Qi et al., 2017; Stohl et al., 2013). Jiao et al. (2014) shows that BC concentrations in snow are poorly correlated with measurements, and a large spread is found  among 25 model simulations, with BC lifetime in the Arctic ranging from about 4 to 23 days, implying large differences in local BC deposition efficiency.

In this study we will focus on the impacts of large-scale atmospheric circulation trends on the transport of BC to the Arctic through a novel statistial methodology, assuming that undersestimating BC concentrations does not significantly affect the spatial distribution of the trends. Three different simulations of the period 1980-2005 are available for our analysis, a reference simulation with annually varying anthropogenic emissions (thereinafter named REF), and two additional simulations with anthropogenic emissions kept constant for the entire period (1980-2005) at the levels of the years 1980 and 2000 (thereinafter named FIX1980 and FIX2000). These three simulations will provide the opportunity to have a preliminary estimate of the uncertainty of our findings due to the different levels and their geographical distribution of anthropogenic emissions used in the different simulations. Other model simulations may be analysed in the future to quantify the uncertainty associated with different chemical mechanisms and physical parameterizations.

## 3 Results

In this study we will focus on the trends of black carbon transport patterns from North Hemisphere mid-latitudes towards the Arctic during winter months. Trends are calculated by the Sen-Kendall method (Sen, 1968). The statistical significance for all trends is set to the 0.05 level and is estimated by the Mann-Kendall test (Mann, 1945). Areas with significant trends are marked with small grey dots in the figures, except for the trends of atmospheric patterns and maximum likelihood estimates of BC that have spatially uniform slopes. We first analyse the trends as simulated by a coupled chemistry-climate model under three different anthropogenic emission scenarios (Section 3.1). This first analysis will provide an estimate of the relative contribution of anthropogenic emissions and natural variability to the transport of BC toward the Arctic. In the second part, through the statistical method described in Section 2, the trend associated with the natural variability are further decoupled into the contributions from the North Hemispheric large-scale atmospheric weather patterns (Section 3.2), which were found to be associated with the observed near surface Arctic warming and sea-ice retreat (Dobricic et al., 2016). In Section 3.3 we will illustrate the total trends due to the three independent atmospheric components and their impact on the inter-annual variability of BC wet deposition and load over the Arctic. The statistical method was applied to a combination of different datasets in order to analyse the spread of different solutions obtained using two different atmospheric reanalysis, three different chemistry-climate model simulations, and for different time periods (Section 3.4).

## 3.1 Total trends of BC simulated by ECHAM5-HAMMOZ

Figure 2 shows the 26-years (1980-2005) trends of BC dry and wet deposition, surface concentration and vertically integrated atmospheric load over to the Arctic for the REF simulation, which includes the effects of both annually varying anthropogenic emissions and inter-annual meteorological variability. Decreasing and statistically significant trends are simulated over almost the entire Arctic for dry and wet deposition, as well as surface concentrations. The main reductions occurred in the Eastern part of the Arctic, from the Scandinavian Peninsula along almost all the Russian northern coastline, indicating the important role of the emission reductions occurred in the last decades in Europe and Russia. Previous studies

also showed that the main transport pathway of air pollution to the Arctic in winter is originating from the Eurasian continent (Sharma et al., 2013; Stohl, 2006). Nevertheless, it is interesting to note that an increasing trend of BC burden is simulated over all Western Arctic and also part of Eastern Siberia. A possible explanation for this result can be found in the increasing anthropogenic emissions in East Asia in the last decades (Figure S1). The long-range transport of aerosol particles from East

Asia to the Arctic occurs in the middle and upper Arctic troposphere, as the warm air masses originated south of the Arctic front are lifted above and affect more the BC burden than surface concentration and deposition, due to slow mixing with surface atmospheric layers. On the contrary the emissions in Eurasia can be often located north of the Artic front during winter and therefore BC is transported in the lower troposphere affecting more surface concentrations and deposition (AMAP, 2015). Similar results were found by Sharma et al. (2013), which estimated three times larger contribution of East

Asia to BC burden in the Arctic between 1990 and 2005, while in the same period the contribution to BC burden and surface concentrations from the FSU declined by 50% and 70%, respectively, The REF, FIX2000 and FIX1980 simulations are driven by the same atmospheric reanalysis and annually varying biomass burning emissions. The inter-annual meteorological variability of the last decades combined with constant anthropogenic emissions (FIX2000 in Figure 3, and FIX1980 in Figure S2) determined a significant increasing trend of BC dry deposition and surface concentrations over the Arctic, BC

wet deposition increased over the Canadian Archipelago and Greenland. Both FIX2000 and FIX1980 simulations show similar patterns, with differences due to the changing geographical distribution of anthropogenic emissions, larger in Europe and North America in 1980, predominant in East Asia in 2000. Both FIX2000 and FIX1980 simulations do not show a significant trend in BC burden over the Arctic (Figures 2d and S2d), which further confirms the role of anthropogenic emissions in East Asia affecting the transport of air pollution at higher altitudes to the Arctic. In some regions, like the

Canadian Archipelago and the western Arctic Ocean, the changes in BC trends due to meteorology and natural variability contribute as much as the changes due to anthropogenic emissions. In the next sections we will further decompose the transport pathways of air pollution to the Arctic associated with the main large scale atmospheric circulation patterns, which are the same that are describing the warming amplification in the polar region and which have characterized the changing climate of the Northern Hemisphere of the last decades.

**3.2 Trends of BC due to atmospheric circulation changes**

Three large scale atmospheric patterns were identified by Dobricic et al. (2016) as closely related to the near surface warming trend in the polar region during winter months (DJF). Two of them, $IC_{NAO}$ and $IC_{SB}$, showed statistically significant trends of surface temperature and geopotential height over the Arctic during the last 36 years (1980-2015), and together could reproduce well the spatial and temporal distribution of the trends found in two atmospheric reanalysis (NCEP and

30 ECMWF ERA-Interim, Section 2.2) In this section we limit our analysis to trends of MLE of BC wet deposition, concentration and load from the FIX2000 simulation and for the longest period available for the construction of coupled anomalies, 26 years (1980-2005). The trends of BC deposition and load are obtained by multiplying the MLE of the BC field by the trend of the associated atmospheric pattern, from the ICA previously performed by Dobricic et al. (2016). We will not

discuss the trends of BC dry deposition as in the ECHAM5-HAMMOZ simulations it is only a small fraction of the total trend of BC deposition in the Arctic (Figures 1b, 2b and S2b).

Figure 1 shows the trends of geopotential height at 850 mbar associated with $IC_{NAO}$ and $IC_{SB}$ patterns estimated by the ICA of NCEP atmospheric reanalysis data by Dobricic et al. (2016), while the arrows visually indicate how these trends may favour certain pollution transport pathways. We can expect that due to the different dynamical structures of $IC_{NAO}$ and $IC_{SB}$ they will differently impact the transport and deposition of pollutants from emission areas in the middle latitudes and their deposition rate once they reach the Arctic atmosphere. The tendency of $IC_{NAO}$ toward the negative phase of the NAO (Figure 1a) forms an anticyclonic anomaly over the large part of the Arctic Ocean and a cyclonic anomaly in the North Atlantic Ocean. The intensity of westerly winds is decreased in the lower troposphere, with lower transport of pollution from North America across the Atlantic Ocean. On the other hand, the $IC_{NAO}$ slightly increases the transport of pollution from northwest America towards the Arctic Ocean. This is also consistent with the results of Hirdman et al. (2010), which found significant correlations between the NAO index and EBC surface concentrations in Alert and Barrow, both in the Western Arctic, with decreasing impact from North Eurasia, and increasing impact from North America. Consistently with the circulation pathways described in Figure 1a, the MLEs of BC wet deposition trends related to $IC_{NAO}$ (Figure 4) show a decreasing trend north of the Eurasian coast and an increasing trend north of America and Greenland. A correlation between the negative phase of the NAO and increasing precipitations and snow accumulation over Western Greenland was also found by previous studies (e.g. Appenzeller et al., 1998; Mosley-Thompson et al., 2005). The BC load has a positive trend over most of the Arctic Ocean, Greenland and the Canadian Archipelago, which may be associated with the dipole of pressure anomalies over the Pacific Ocean which is also favouring the export of polluted air masses from East Asia into North America and the Arctic (Figure 1a). Sharma et al. (2013) previously showed that the contribution of East Asian BC emissions in the Arctic above 200 mb is the largest. The trends of average BC concentrations at different altitudes above 60°N at four different longitudinal portions (quadrants) of the Arctic (0°-90°E; 90°-180°E; 180°W-90°W; 90°W-0°) are shown in Figure 5. In all the four Arctic quadrants an increasing trend of BC concentrations is estimated at an altitude of about 10 km, near the tropopause. Above the tropopause a shift in the vertical profile of BC concentration is observed, with decreasing trend between the tropopause and 15 km, and increasing trend above 15 km and 70°N. As shown in Dobricic et al. (2016), the evolution of the $IC_{NAO}$ pattern from December to February is indicating a weakening of the stratospheric vortex linking the tropospheric perturbation with the stratosphere, in agreement with findings by Feldstein and Lee (2014). There is an enhanced mixing anomaly between the troposphere and the stratosphere indicated by increased BC concentrations above and decreased below the tropopause. Near the surface the $IC_{NAO}$ has a small impact on BC concentrations with few differences in the four Arctic quadrants. Only between 0°-90°E there is a small increasing trend above 70°N, while between 90°E-0° the BC concentrations increase only below 70°N.

The $IC_{SB}$ pattern (Figure 1b), similarly to a Scandinavia Blocking, consists of an anticyclonic centre near Scandinavia and weaker centres of opposite sign over South Western Europe and Siberia/Mongolia. In this case advection changes in the lower troposphere indicated by the arrows favour the larger transport of pollutants from Europe directly to the sea ice

covered areas of the Arctic Ocean, while due to the southward position of the anticyclonic anomaly the vertical stability over the Arctic is not changed significantly. The large high pressure anomaly over the North Pacific Ocean decreases the Asian influence on the Arctic. In a dynamically consistent way it advects the warm air from the south to the Barents Sea and the cold air from the Arctic into Siberia. This circulation pattern results in an increased transport of pollution from Central Europe directly to the Barents Sea and the Arctic, while the transport from Siberia to the Arctic is suppressed. All estimates of the deposition of BC related to $IC_{SB}$ show a strong positive trend that extends from the Fram Strait, the Barents and Kara Seas to the central part of the Arctic Ocean (Figure 4). The positive trend of the load spreads over the Norwegian Sea and Scandinavia and it indicates the northward advection of the pollution from Europe towards the Arctic Ocean. The relationship between $IC_{SB}$ deposition and load trends and the trend of the anticyclonic circulation over Scandinavia is especially evident in February (not shown) when the Scandinavian anticyclone trend is the most pronounced (Dobricic et al., 2016). The vertical structure of the estimated BC concentration trend (Figure 6) shows that in this conditions the maximum concentration trend is located in the lower troposphere below 3 km and mainly in the first quadrant of the Arctic (0°-90°E). A strong increasing trend is extending from mid latitudes up to 90°N near the surface. However a negative trend is observed over the entire Arctic near the tropopause and above. The $IC_{SB}$ pattern is associated with a negative phase of the ENSO (La Niña), with colder surface temperature in the equatorial Pacific. The opposite signs of perturbations in the equatorial Pacific and in Scandinavia maintain a planetary wave trapped in the troposphere and compared to the $IC_{NAO}$ pattern a lower transport to the stratosphere (Dobricic et al., 2016).

### 3.3 Total trends of BC Maximum Likelihood Estimates

Figure 7 shows the reconstructed total trends of BC wet deposition and load, as the sum of the winter seasonal mean of maximum likelihood estimates associated with three independent atmospheric patterns ($IC_{NAO}$, $IC_{SB}$ and $IC_{ENSO}$) multiplied by their temporal trends. The BC wet deposition in winter increased in the last decades due to changing atmospheric circulation over a large part of the polar region. The largest increases were estimated over the Barents Sea, the Kara Sea and the Fram strait, south Greenland, Alaska and the Canadian Archipelago, but the increasing deposition trend extends over most of the Arctic Ocean covered by sea-ice in winter (the grey line in Figure 7 represents the limit of sea-ice and snow covered areas averaged since the 1980s). Decreasing BC wet deposition is estimated over most of Siberia and the Laptev Sea and East Siberian Sea. The trend of atmospheric load of BC reconstructed by the three independent components shows a net increasing trend over the Arctic north of about 75°N and the decreasing trend over the mid latitudes, except over west Russia, the Scandinavian Peninsula and the Norwegian Sea. The trends reconstructed by the MLE analysis are similar to the trends simulated by the FIX2000 chemistry-climate model simulation (Figure 3a,d). In particular wet deposition trends are well represented in terms of both magnitude and spatial distribution. The trend of BC load in the FIX2000 simulation is not significant over the polar region, on the other hand part of the noise was removed from the coupled atmospheric-chemistry anomalies by performing and EOF transformation and retaining only the components with largest eigenvalues.

The temporal variability of the BC deposition anomalies in the period 1980-2015 may be estimated for any area and for each pattern. In particular it is interesting to evaluate the combined and individual impacts of $IC_{NAO}$, $IC_{SB}$ and $IC_{ENSO}$ patterns on the total BC wet deposition and load over the Arctic Ocean, between 80°N-90°N. Figure 8 shows that the $IC_{NAO}$ pattern produces deposition anomalies with a smaller inter-annual variability than those driven by the $IC_{SB}$ pattern. The contribution of $IC_{ENSO}$ to the total deposition is relevant only in the third quadrant of the Arctic (180°W-90°W), north of the Canadian Archipelago. The largest variability is found in the first quadrant, north of Barents and Kara Seas, up to a factor of 10 larger than the other sectors of the Arctic. Deposition anomalies from the $IC_{NAO}$ pattern are mostly negative till the year 2000, and after become positive. Those from the $IC_{SB}$ pattern show a strong increase after the year 2000 that dominates the whole trend from 1980 to 2015, which corresponds well to an acceleration in the winter sea ice decrease rate over the Barents Sea, which is often coupled with the formation of the anticyclonic anomaly over Scandinavia (Cohen et al., 2014; Dobricic et al., 2016; Sato et al., 2014; Screen et al., 2013). Figure 9 shows the temporal contribution of $IC_{NAO}$, $IC_{SB}$ and $IC_{ENSO}$ patterns to the total BC load over the Arctic quadrants. In this case, the $IC_{NAO}$ pattern drives the inter-annual variability, with negative values until the late 1990s and positive in the last 15 years. The $IC_{NAO}$ pattern affects BC load over the entire Arctic in a similar way, while the contribution of the $IC_{SB}$ pattern has an opposite trend except for the first quadrant, where it is only a small fraction of the total variability. As discussed in the previous section, the $IC_{SB}$ pattern increase the transport of BC directly from Central Europe to the Barents and Kara Sea, while it decreases the transport from Siberia. The $IC_{SB}$ pattern has a negative contribution in the last 10 years in the other parts of the Arctic, and in this regions it can be as large as the $IC_{NAO}$ contribution. In Figure 8 and Figure 9 we can see that also the temporal variability is partially reconstructed by the applied Bayesian statistical approach. The correlation coefficients between the inter-annual variability reconstructed by the MLE and the FIX2000 model simulation for the common period 1980-2005, range between 0.35 and 0.4 for BC wet deposition and between 0.14 and 0.41 for BC load over the different sectors of the Arctic.

**3.4 Uncertainty of the BC Maximum Likelihood Estimates**

In order to test the robustness of the statistical method, we have estimated the MLE of pollutant distributions for a set of 8 combinations of coupled atmospheric reanalysis and BC simulations (Table 1). Dobricic et al. (2016) applied the ICA to two different atmospheric reanalysis of the period 1980-2015, NCEP and ERA-INTERIM, and similar atmospheric patterns and trends of the independent components were found. A first set of MLEs combined the atmospheric patterns computed from NCEP and ERA-INTERIM with two global chemistry-climate simulations, which used constant anthropogenic emissions (FIX2000, and FIX1980, see Section 2.3). The MLEs were computed using the simulated pollutant fields for the longest period available, 26 years from 1980 to 2005. The results from NCEP-2000-A have been explained in details in the previous sections.

Figure 10 shows the total trends ($IC_{NAO}+IC_{SB}+IC_{ENSO}$) of BC wet deposition estimated using the FIX2000 and FIX1980 simulations to build the pollutants and atmospheric coupled anomalies. All the 4 estimates are consistent in terms of geographical distribution, magnitude of the estimated trends and their statistical significance. Consistent results are obtained

also for the total trends of BC load (Figure S3 in the supplementary material). Only small differences are due to the two atmospheric reanalysis and the two atmospheric composition simulations, which have a different geographical distribution of anthropogenic emissions (Figure S1).

In a second set of MLEs, we used the REF model simulation, which includes annually varying anthropogenic emissions. In this case, the sudden drop of anthropogenic emissions in Eastern Europe and Russia, and increase in East Asia in the early 1990s (Figure S1d), introduced a strong discontinuous change in the transport variability that was impossible to resolve by the statistical method. Thus we chose a shorter period to form the simulated chemistry and atmospheric coupled anomalies, 13 years (1993-2005) after the drop of anthropogenic emissions. Figure 11 shows the estimated trends of BC wet deposition (trends of BC load are shown in Figure S4 of the supplementary material) using only 13 years of coupled anomalies, from the REF simulation and also from the FIX2000 simulation for comparison. Also in this case the results shown in Figure 10 and Figure 11 are consistent, using a shorter period to form the simulated coupled anomalies and with both constant and varying anthropogenic emissions (without large emission changes in a short time). In the most remote Arctic region covered by sea-ice, far from the main anthropogenic sources, the entire set of estimated trends of BC transport and deposition are very close to each other, meaning that the applied methodology can robustly approximate the transport of pollutants to this region and the trends associated with different large scale circulation patterns.

## 4 Conclusions

The feedbacks between the global warming and arctic amplification with sea-ice retreat and impacts on large-scale atmospheric circulation are still contradictory. The response of mid-latitude weather to the Arctic warming and sea-ice cover changes of the last decades is highly uncertain due to nonlinear processes involved in the Arctic and subarctic climate system (Overland et al., 2016). Some studies find only weak or non-existent relationships between mid-latitude weather structures and Arctic warming (e.g. Screen and Simmonds, 2013; Barnes et al., 2014), while others found correlations between sea-ice retreat in winter over the Barents and Kara Seas and hemispheric scale impacts (e.g. Deser et al., 2007; Petoukhov and Semenov, 2010; Screen et al., 2013; Mori et al., 2014; Di Capua and Coumou, 2016). Recently, The trends from two atmospheric reanalysis (Figure 2 and Figure S3 of Dobricic et al., 2016) of winter near surface warming of the Northern Hemisphere are approximated by three independent components which are also very similar to well-known atmospheric oscillations, the NAO, SB, and ENSO. We developed a statistical algorithm to combine the estimates of these independent components ($IC_{NAO}$, $IC_{SB}$, and $IC_{ENSO}$) with chemistry-climate model simulations of transport and deposition of black carbon (BC) over the Arctic. In this way we estimate how changes in the large-scale atmospheric circulation impact the BC transport and deposition independently of the changes in emissions. For a given independent component of the atmospheric variability our algorithm estimates the most likelihood deposition and load of BC consistent with the variability of the atmosphere-chemistry coupled anomalies. To test the robustness of the method and the spread of the possible solutions, the method is applied with two atmospheric reanalysis, NCEP and ERA-INTERIM, and three realizations of

coupled anomalies from ECHAM5-HAMMOZ global model simulations, with constant and annually varying anthropogenic emissions.

The main results are summarized in the following points:

- The trends of BC concentrations and deposition fluxes simulated by a global chemistry-climate model (ECHAM5-HAMMOZ) indicated a strong reduction of near surface transport and deposition to the Arctic due to anthropogenic emission reductions in Eurasia, in agreement with observations and other modelling studies (Gong et al., 2010; Hirdman et al., 2010), and an increasing BC load mainly due to the emissions from East Asia (Sharma et al., 2013). On the other hand model simulations with constant anthropogenic emissions indicated how the effect of changing meteorology in the last decades may have also determined significant trends of BC transport to the Arctic;

- The increasing trend of the anticyclonic circulation anomaly over the Arctic ($IC_{NAO}$) determines a decrease of the pollution transport pathway from Eurasia to the Arctic, with a decrease in BC surface concentrations and deposition particularly in the Eastern Arctic. On the other hand, the $IC_{NAO}$ trend favours a stable vertical stratification which determines an increase of BC transport in the upper troposphere from East Asia and North America with increasing trend of total column BC concentrations over the entire Arctic Ocean;

- A second component, named $IC_{SB}$, directly transport BC in the Arctic lower troposphere from central Europe. The increasing trend of an anticyclonic anomaly over Scandinavia and West Siberia favours the transport of air pollution directly from Europe to the Barents Sea with a significant increase of BC wet deposition over the entire Arctic Ocean. Also this pattern determines a decrease of pollution transport from East Eurasia to the Arctic and an overall decrease of BC load over the entire Arctic upper troposphere, except over the Barents Sea where the surface concentration trends is strongly increasing;

- A third component, which is more related to anomalies in the tropical Pacific Ocean and named $IC_{ENSO}$, has a smaller and statistically less significant impact on the transport and deposition of BC to the Arctic;

- The combined impact of all three atmospheric patterns on BC wet deposition determines a significant increasing trend between 1980 and 2015 over almost the entire Arctic Ocean, the Barents Sea and Fram Strait, Greenland, Alaska and the Canadian Archipelago. A small significant decreasing trend is estimated the East Siberian and Laptev Seas. The BC load increases over the whole Arctic Ocean, with a significant trend only North of Greenland and the Barents Sea;

- The estimated inter-annual variability and trend of BC wet deposition over the Arctic Ocean is mainly driven by the $IC_{SB}$ pattern, with positive anomalies persisting for the last 10 years, 2005-2015. Smaller contributions are estimated for $IC_{ENSO}$, only between 180°W-90°W, and NAO 90°W-0°. The inter-annual variability and trend of BC load over the Arctic is mainly driven by the $IC_{NAO}$ atmospheric pattern, in

particular north of the Barents Sea. In the other sectors of the Arctic, also $IC_{SB}$ contributes to the load total variability.

- We have shown the robustness of a Bayesian approach to estimate the maximum likelihood distribution and trends of air pollutants associate with specific atmospheric patterns. The estimates obtained from different atmospheric reanalysis and global chemistry-climate model simulations are consistent. Problems may arise to resolve the total variability in presence of strong emission discontinuous changes. Although it was possible to separate the effects of changing anthropogenic emissions from the meteorological variability using only global chemistry-climate simulations, by MLE further provided information on the feedback impacts of single atmospheric processes related to global warming and se ice retreat on pollution transport. By using outputs from existing coupled model simulations the method may be applied with other atmospheric processes related to the pollution transport and to other temporal intervals.

Our results show how in winter atmospheric processes linked to enhanced Arctic warming and sea-ice retreat may impact the BC deposition on the sea ice itself. BC deposition is mainly decreasing over the area that in the recent years is characterized by the thin first-year sea ice, while it is increasing over the area with the thicker multiyear sea ice, which is becoming a smaller fraction of the total sea ice cover in last year's summers. During the winter polar night the deposited BC has no impact on the sea ice melting. When the sun appears in the spring the short wave radiation starts to melt the fresh snow deposited on the sea ice. The BC impact on the albedo may be the most important at the end of the winter season, just before the snow starts to melt and surface becomes darker. Therefore, the BC deposition in winter may accelerate the initial melting of the snow and by an integrated effect it may result in the increased melting at the end of summer. The first-year sea ice may be completely melted every summer and the decrease of the BC deposition might not be the most important factor controlling its coverage in the late summer. On the other hand, with the increased deposition of BC the multiyear sea ice may melt more and its thickness may become reduced with respect to the previous year. If this process happens consistently over several decades it might significantly contribute to reduction of the thickness of the multiyear sea ice. Our estimate of the temporal variability of the BC deposition over the multiyear sea ice indicates that the deposition in the winter due to the sea ice retreat feedback increased especially in the last ten years.

The increasing trend of the BC deposition due to changes in atmospheric circulation emphasizes the importance of the reduction of BC anthropogenic emissions in the mid latitudes. In particular the $IC_{SB}$ pattern favours the transport to the Arctic Ocean of polluted air masses from Europe, and increased deposition on the multiyear sea ice. Policies that reduced anthropogenic emissions in the last decades, therefore, reduced the risk of a further acceleration of the melting of the multiyear sea ice. A strong shift of the anthropogenic emissions from North America and Europe to East and South Asia also contributed to reduce the transport of BC to the Arctic from Eurasia, which represents the main transport pathway. We may have an indication of the impact of this geographical emissions shift by comparing the trends estimated using the FIX2000 and FIX1980 BC simulations (Figure S5). The BC deposition trend increased over the East Siberian and Laptev Sea considering the anthropogenic emissions of year 2000 compared to 1980, indicating that the contribution from the Asian

emissions may play an important role in the future (e.g. Sand et al., 2015). This increase is affecting deeper into the Arctic Ocean in particular in association with the $IC_{NAO}$ pattern. A similar impact of the increasing emissions in Asia was estimated for the BC burden, in particular associated with $IC_{NAO}$ and extending from Siberia, across the Arctic Ocean up to the Canadian Archipelago and Greenland.

In order to further understand the link between the global warming and the trend of BC transport and deposition will be necessary to extend the study to other seasons. In particular the major precipitation and wet deposition of BC happen in the summer when also the reduction of the ice cover in the recent years is the largest. Furthermore, although the use of different estimates for atmospheric patterns and coupled atmosphere-chemistry relationships provides some information on their uncertainty, in the future the same method may be applied with several coupled models and multi-year observations in the

Arctic to further assess the uncertainties. The same methodology may be also applied to estimate trends of BC transport to the Arctic associated with atmospheric circulation changes in the future, for example using the global climate simulations instead of atmospheric reanalysis.

**Acknowledgements**

We thank the anonymous reviewers for the helpful comments and suggestions during the review phase.

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

**Table 1: Combination of atmospheric reanalysis, global chemistry-climate model simulations and time periods used to generate MLE of BC deposition and load.**

| Estimate name | Atmospheric reanalysis | BC simulation | Period |
|---|---|---|---|
| NCEP-2000-A | NCEP | FIX2000 | 1980-2005 |
| ERA-2000-A | ERA-Interim | FIX2000 | 1980-2005 |
| NCEP-1980-A | NCEP | FIX1980 | 1980-2005 |
| ERA-1980-A | ERA-Interim | FIX1980 | 1980-2005 |
| NCEP-REF-B | NCEP | REF | 1993-2005 |
| ERA-REF-B | ERA-Interim | REF | 1993-2005 |
| NCEP-2000-B | NCEP | FIX2000 | 1993-2005 |
| ERA-2000-B | ERA-Interim | FIX2000 | 1993-2005 |

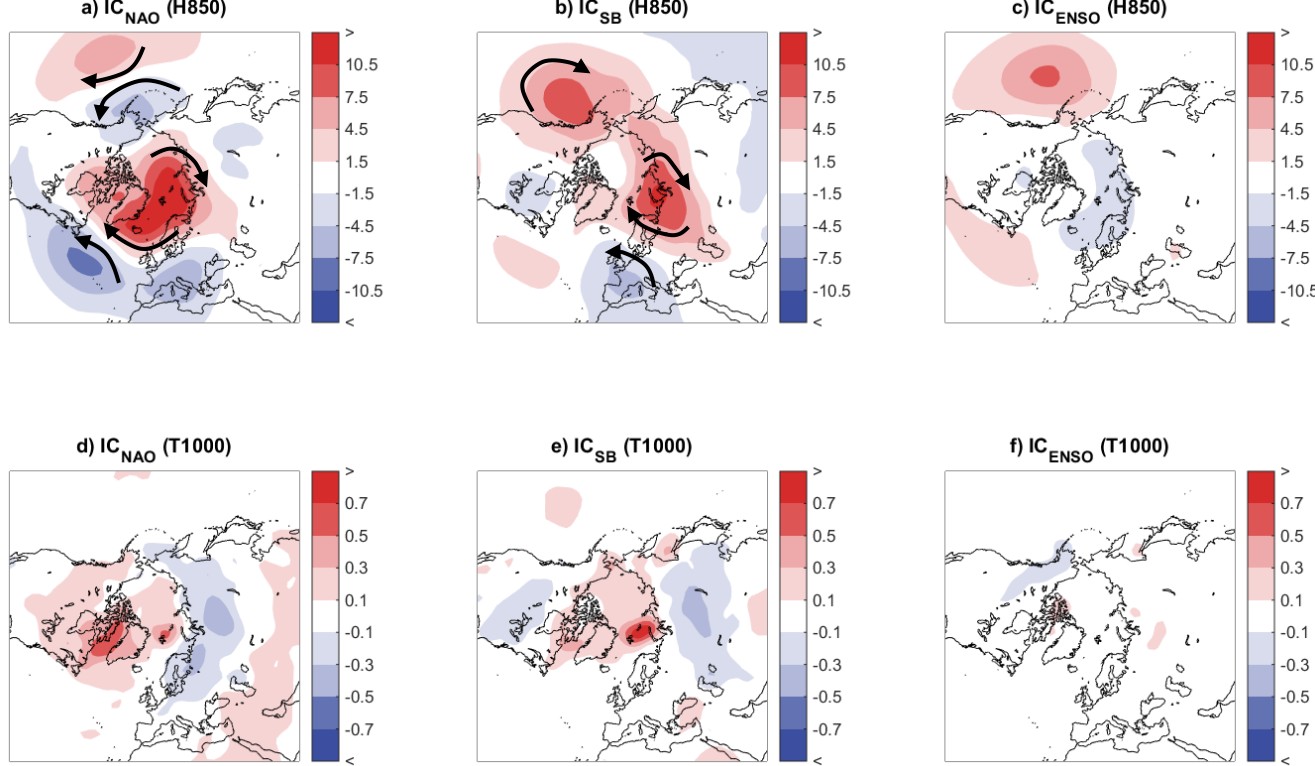

**Figure 1** Winter trends (1980-2015) of geopotential height at 850 mbar (H850, m year$^{-1}$, top) and near surface temperature at 1000 mbar (T1000 K year$^{-1}$, bottom) of the independent atmospheric patterns related to North Atlantic Oscillation (IC$_{NAO}$), Scandinavian Blocking (IC$_{SB}$), and El Nino-Southern Oscillation (IC$_{ENSO}$). The arrows provide a qualitative representation of the circulation paths tendencies associated with the a) IC$_{NAO}$ and b) IC$_{SB}$ independent component patterns.

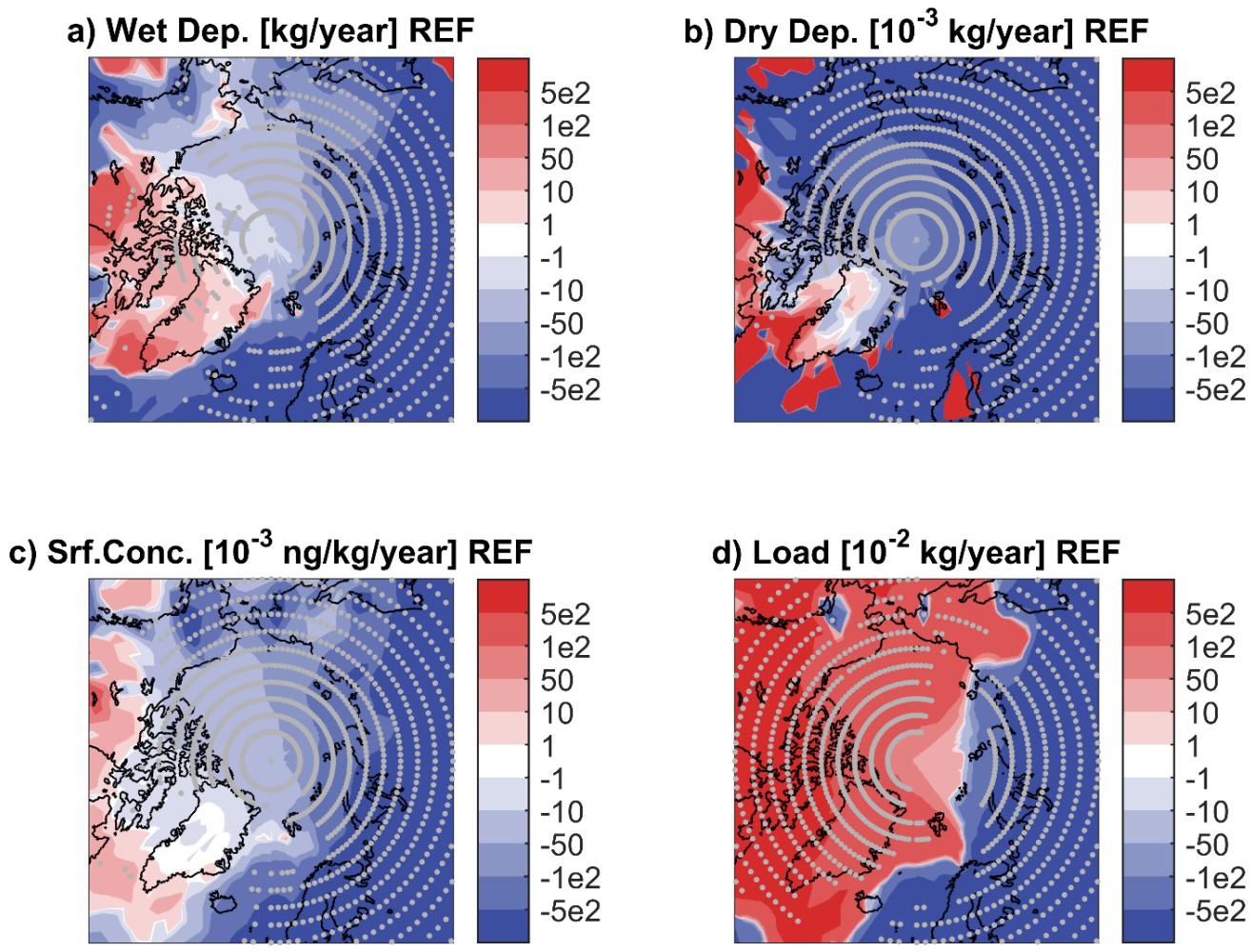

**Figure 2 Winter (DJF) trends (1980-2005) of BC wet deposition (a), dry deposition (b), surface concentration (d), and total load (d) for the ECHAM5-HAMMOZ REF simulation. Grey dots represent the grid points with trend significant at 5% level.**

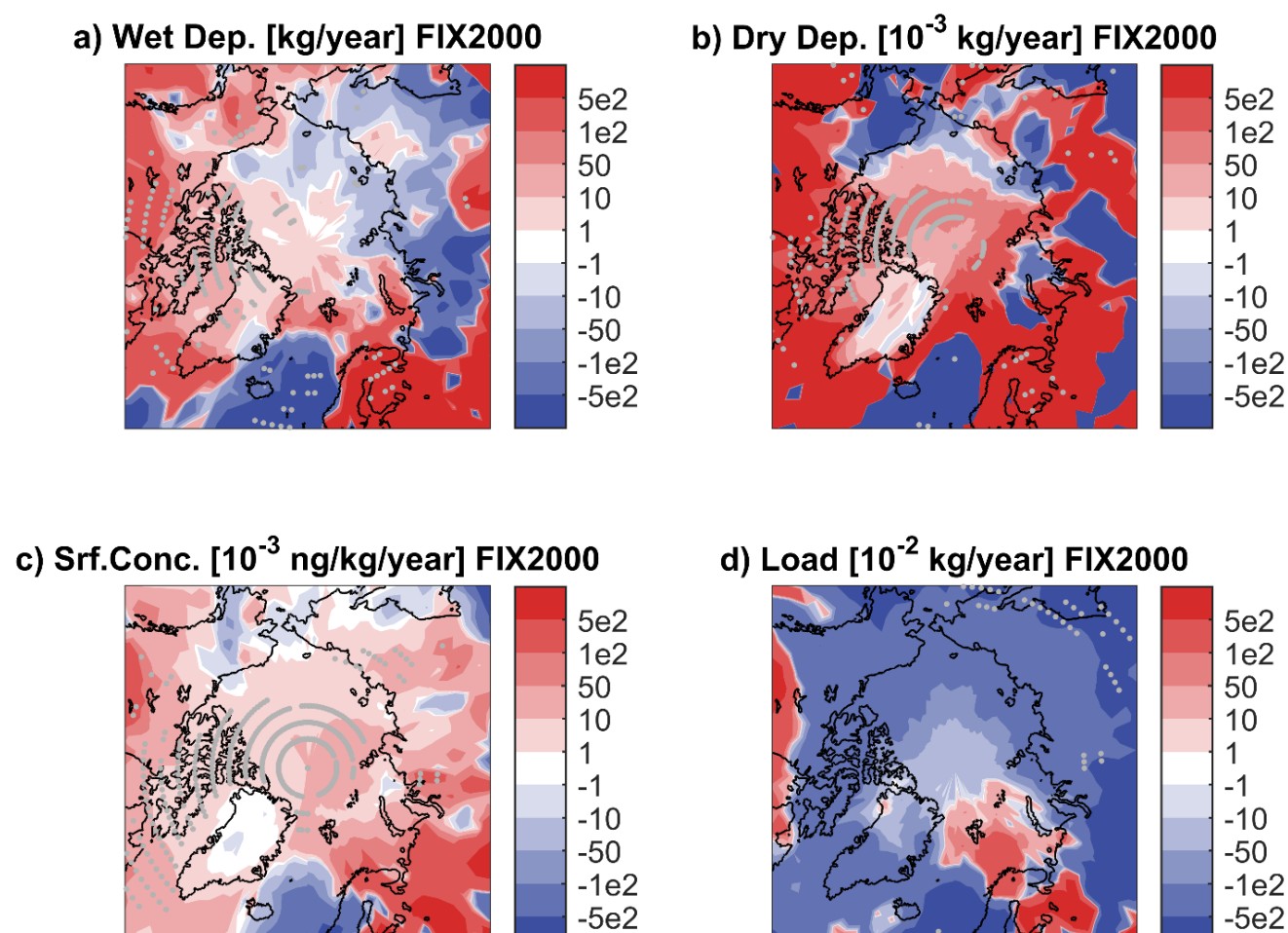

**Figure 3 Winter (DJF) trends (1980-2005) of BC wet deposition (a), dry deposition (b), surface concentration (d), and total load (d) for the ECHAM5-HAMMOZ FIX2000 simulation. Grey dots represent the grid points with trend significant at 5% level.**

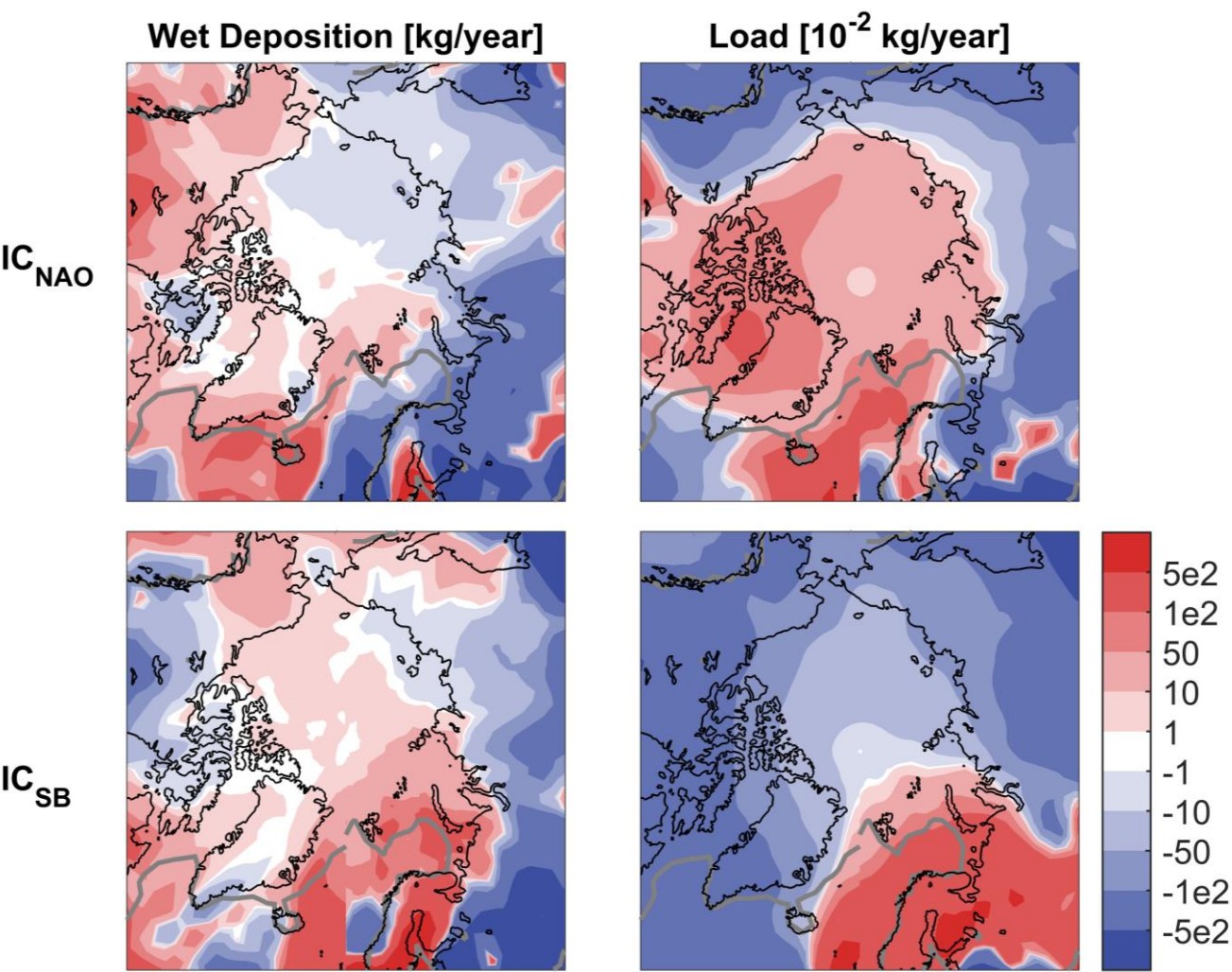

**Figure 4 Winter (DJF) trends (1980-2015) of maximum likelihood estimates (MLE) of BC wet deposition (kg/year) and load (kg x $10^{-2}$/year) associated with $IC_{NAO}$ and $IC_{SB}$. The grey line represents the mean winter sea ice and snow cover larger than 50% since 1980 to now. NAO and SB trends are both significant at 5% level.**

## Trends of BC concentrations due to IC$_{NAO}$

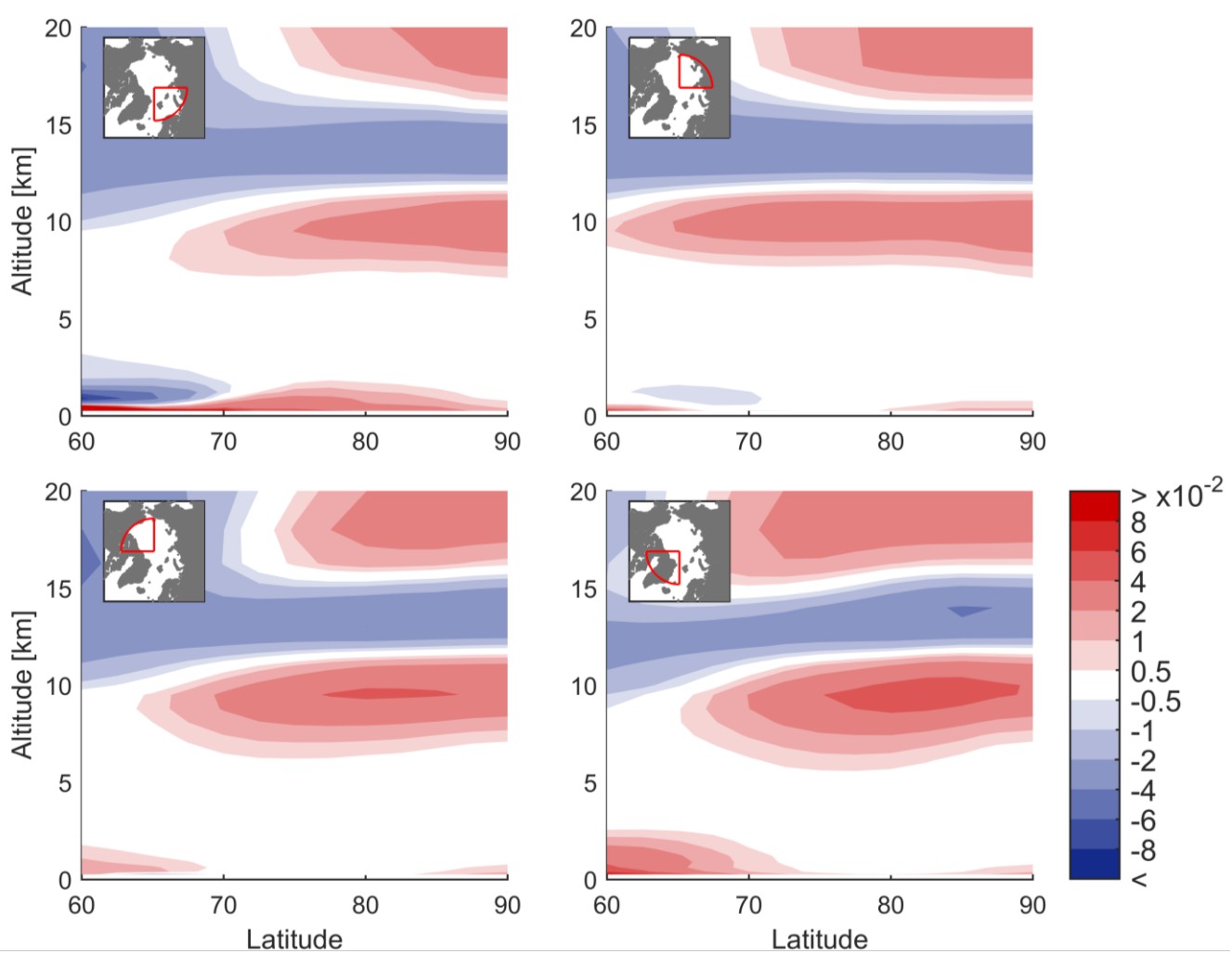

**Figure 5 Winter (DJF) trends (1980-2015) of maximum likelihood estimates of BC concentrations (ng/kg/year) associated with IC$_{NAO}$ circulation pattern. BC concentrations at each model vertical level and latitudes above 60°N are averaged over 90° longitudes segments of the Arctic, indicated in the map on the left upper corner of each plot.**

## Trends of BC concentrations due to IC$_{SB}$

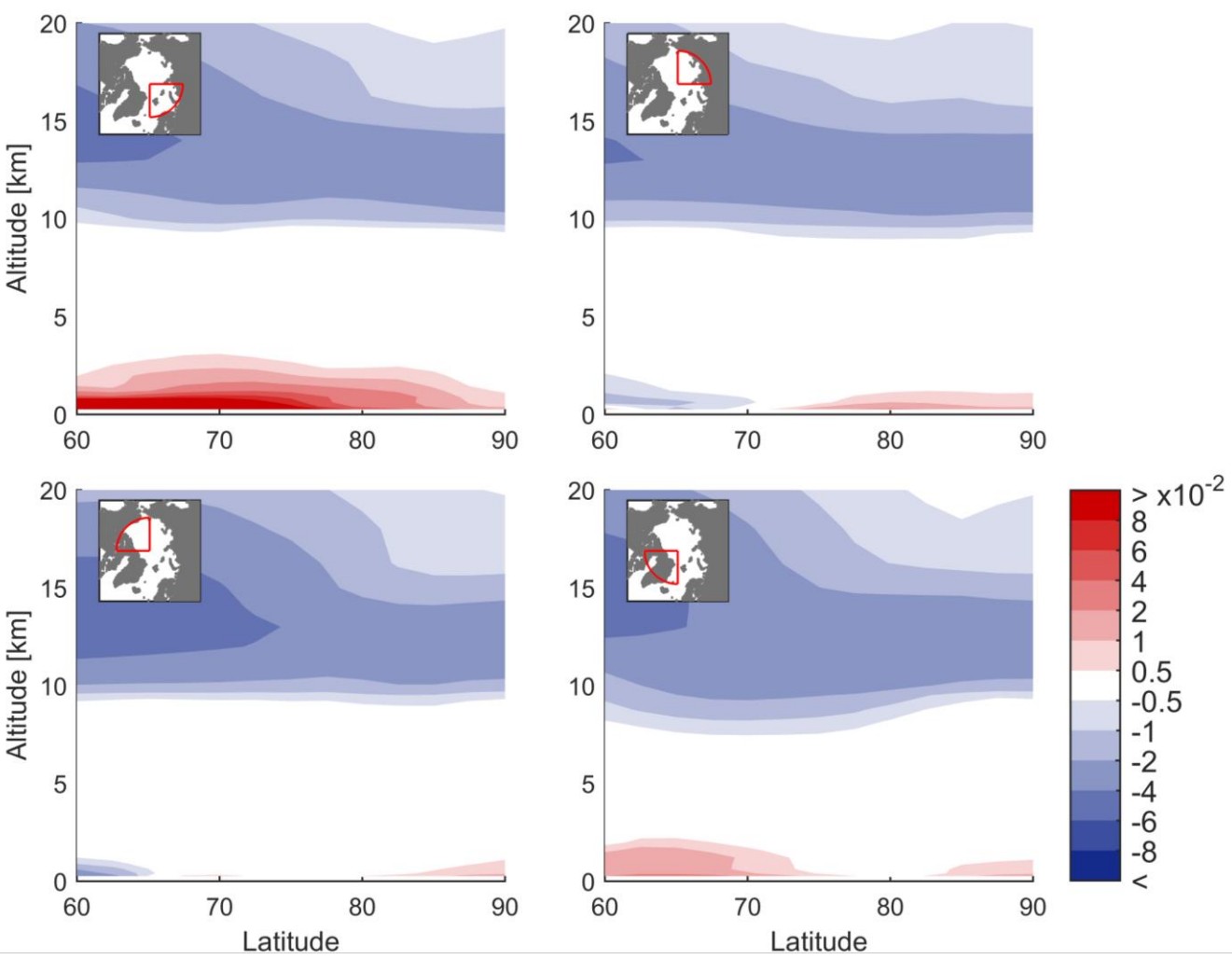

**Figure 6 Winter (DJF) trends (1980-2005) of maximum likelihood estimates of BC concentrations (ng/kg/year) associated with IC$_{SB}$ circulation pattern. BC concentrations at each model vertical level and latitudes above 60°N are averaged over 90° longitudes segments of the Arctic, indicated in the map on the left upper corner of each plot.**

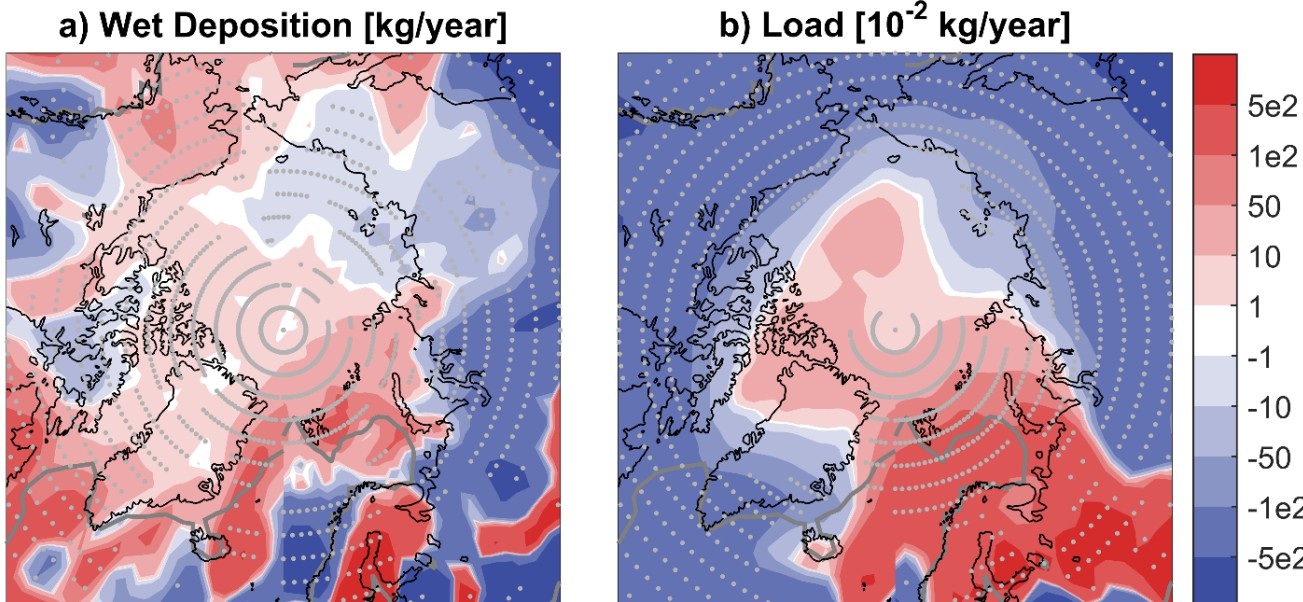

**Figure 7 Total winter (DJF) trends (1980-2015) of maximum likelihood estimates (MLE) of BC wet deposition (kg/year) and load (kg x 10^-2/year) associated with three atmospheric circulation patterns (Total=IC_NAO+IC_SB+IC_ENSO). The grey line represents the mean winter sea ice and snow cover larger than 50% since 1980 to now. Grey dots represent the grid points with trend significant at 5% level.**

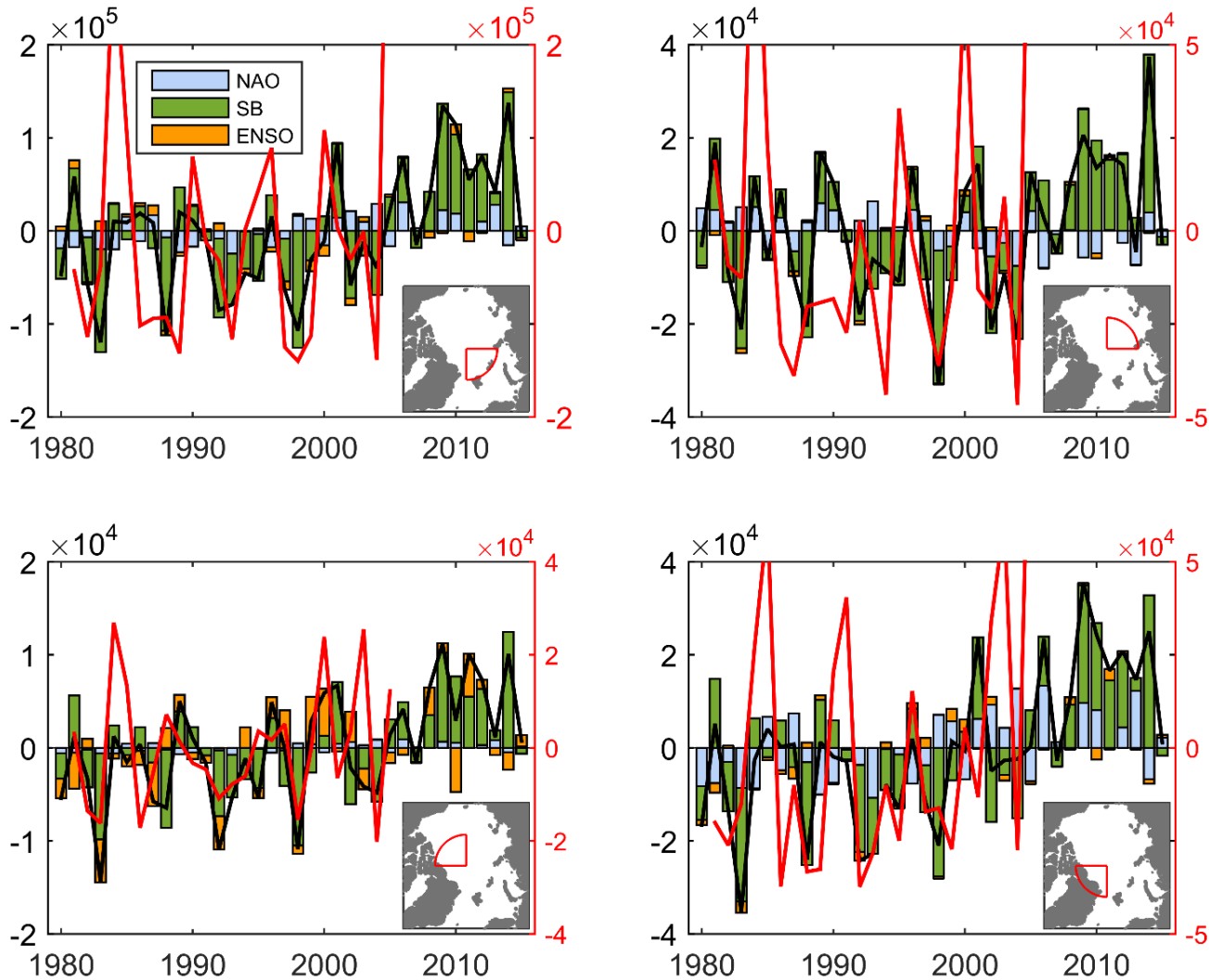

**Figure 8 Temporal variability of total BC wet deposition anomaly over 90° longitude segments of the Arctic and between 80°N-90°N (kg/year) related to the circulation patterns $IC_{NAO}$, $IC_{SB}$ and $IC_{ENSO}$. The black line is the sum of the three components, the red line (right y-axis) is the total BC wet deposition anomaly from the FIX2000 simulation.**

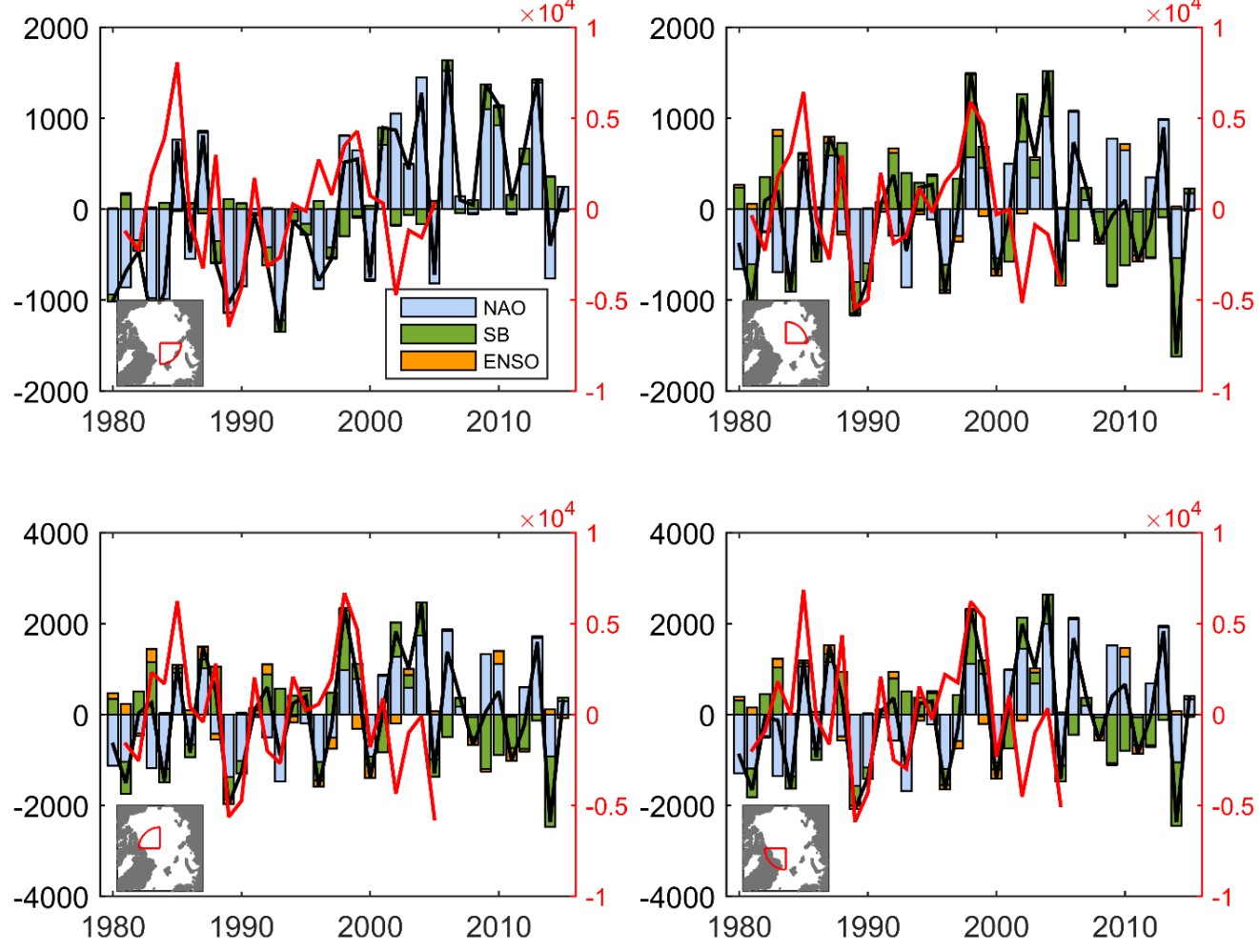

**Figure 9 Temporal variability of total BC load anomaly over 90° longitude segments of the Arctic and between 80°N-90°N (kg/year) related to the circulation patterns IC$_{NAO}$, IC$_{SB}$ and IC$_{ENSO}$. The black line is the sum of the three components, the red line (right y-axis) is the total BC load anomaly from the FIX2000 simulation.**

# BC Wet Deposition [kg/year]

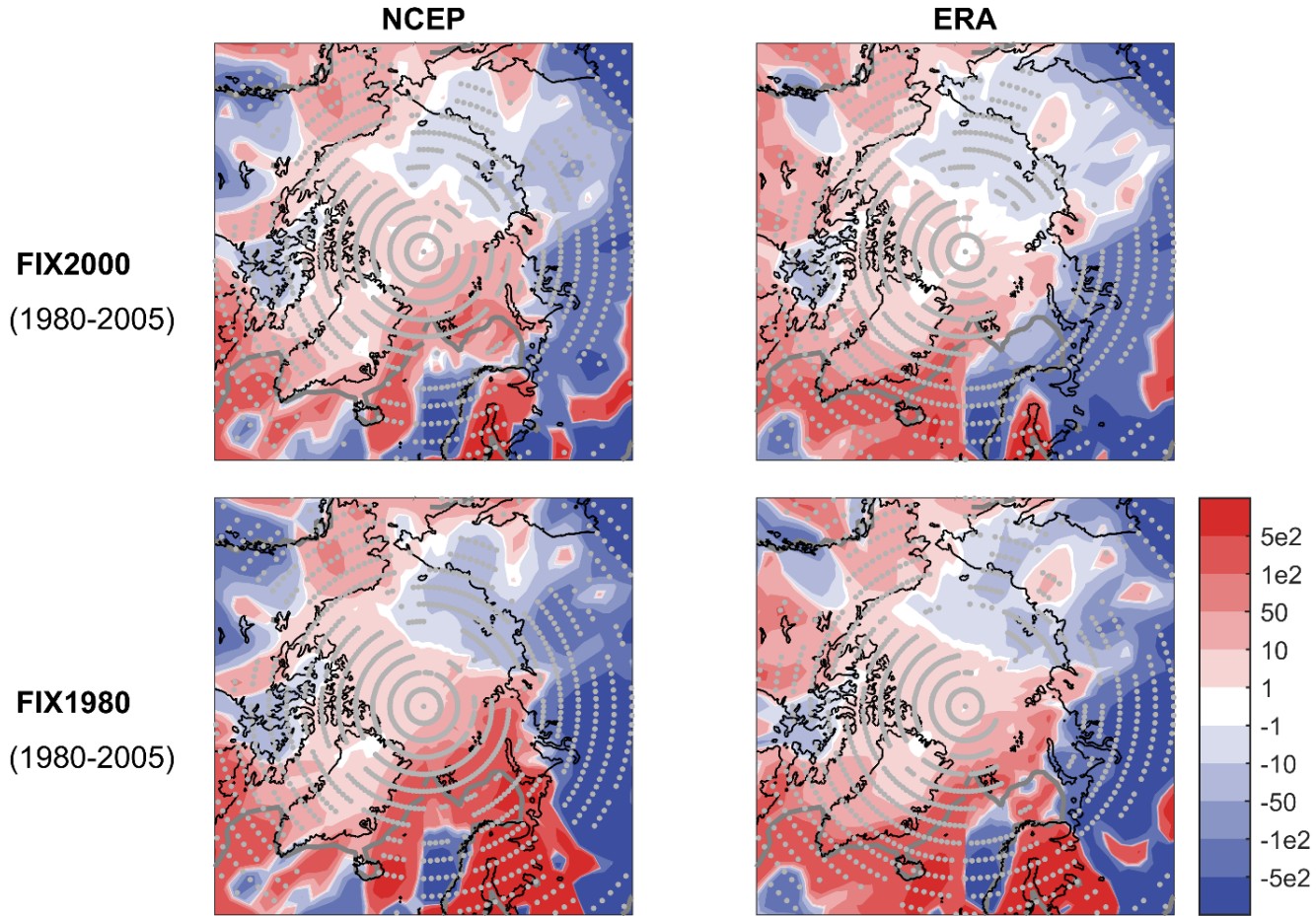

**Figure 10** Four different estimates (see also Table 1) of total trends of maximum likelihood BC wet deposition (kg/year) associated with three atmospheric circulation patterns (Total=NAO+SB+ENSO). The grey line represents the mean winter sea ice and snow cover larger than 50% since 1980 to now. Grey dots represent the grid points with trend significant at 5% level.

# BC Wet Deposition [kg/year]

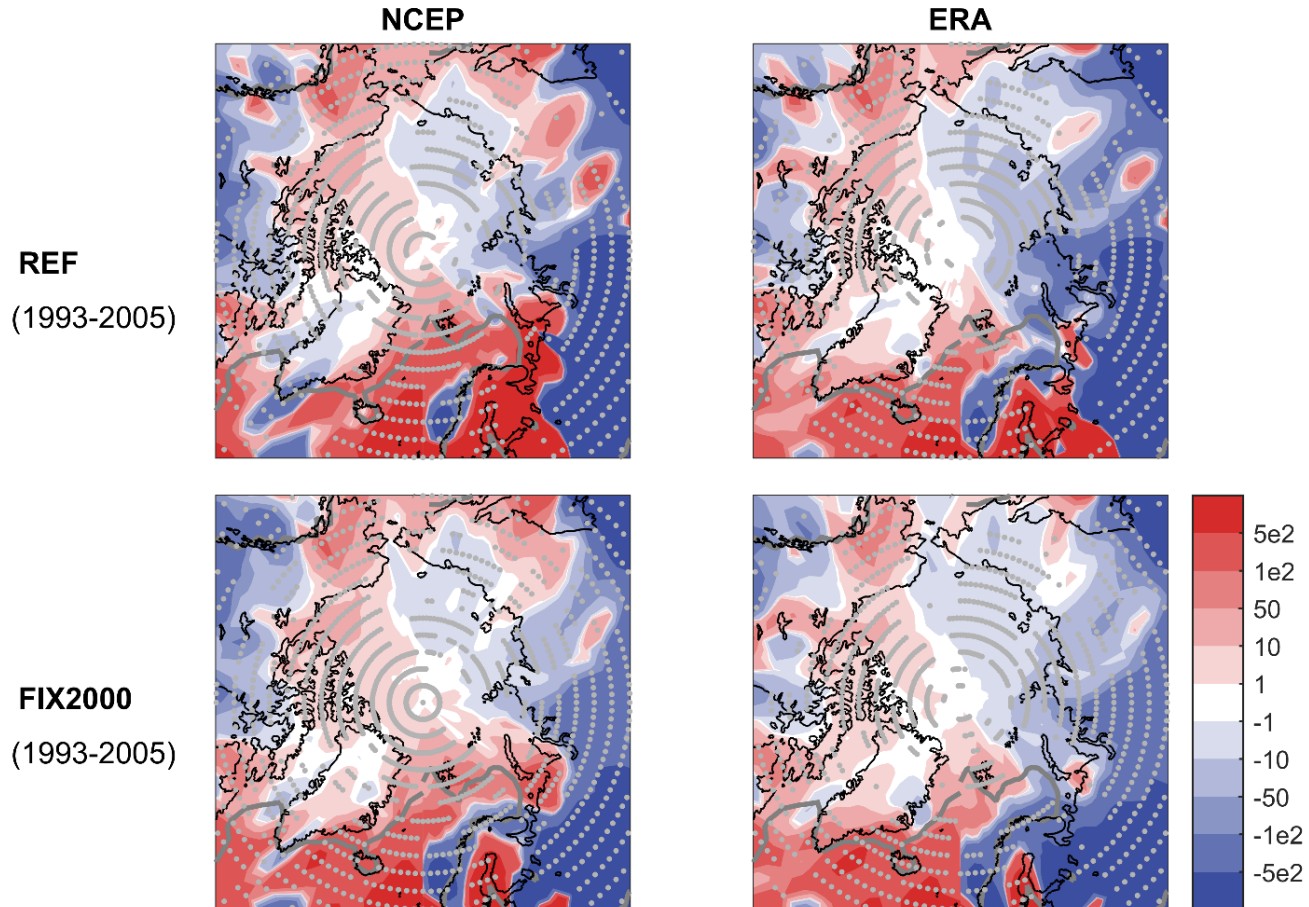

**Figure 11** Four different estimates (see also Table 1) of total trends of maximum likelihood BC wet deposition (kg/year) associated with three atmospheric circulation patterns (Total=NAO+SB+ENSO). The grey line represents the mean winter sea ice and snow cover larger than 50% since 1980 to now. Grey dots represent the grid points with trend significant at 5% level.