# Peer review of "Impacts of large-scale atmospheric circulation changes in winter on Black Carbon transport and deposition to the Arctic"

_Atmospheric Chemistry and Physics, 2016_

## Referee Comment (RC1) · Anonymous Referee #1 · 27 Jan 2017

Pozzoli et al. investigate how changes in three circulation patterns defined in another study influence the transport of black carbon aerosols to the Arctic and the deposition there. The topic is interesting and within the scope of ACP. While the statistical methods the authors use to link the transport of BC with circulation patterns seems valid and well-explained, I am concerned about how the study is presented in general. The authors conclude that changes in atmospheric circulation patterns due to Arctic warming and reduced sea-ice has significantly impacted the BC transport and deposition. For instance, the authors argue that increased blocking over Scandinavia has led to an increase in deposition in West Arctic. The link between Arctic climate change and weather at mid latitudes is controversial, and the authors do not show any attempt to

discuss this at all. There is hardly any comparison with previous work or with observations. Therefore, I cannot recommend publication of this manuscript in its current form. Below are some recommendations for improvements.

General comments:

1. I think you need to explain the method in Dobricic et al. 2016 in a bit more detail in terms of correlations factors and eigenvalues, as this study is important for your results. How much of the wintertime variability do the patterns explain? How are these patterns linked to sea-ice melting? For instance, you say that: 'The relationship between NAO and SB and the sea ice melting over the Arctic is visible in near-surface temperature trends, the associated warming over the Arctic is the largest over the Barents Sea and west of Greenland.' There is a relationship because you see similar geographical patterns?

2. I think a discussion of the uncertainties using a 'simple' cause-and-effect approach should be included here. The Arctic climate system is complex and noisy, and relationships between e.g. sea-ice retreat and circulation patterns are nonlinear. Many processes may mask each other out. A new perspective paper in Nature Climate Change highlights some of these challenges, and argue that a single cause is unlikely: http://www.nature.com/nclimate/journal/v6/n11/pdf/nclimate3121.pdf

3. In general, I think this manuscript should include more references to previous work.

4. Your conclusions differ from other studies. Hirdman et al. 2010 as you mention in the introduction, finds that changes in emissions were dominant, and that circulation changes only explained a minor fraction of the observed trend. How is your study different? And why do you think they differ so much? Please put your work into context.

5. There are no comparisons with observations at all. The observations in the Arctic are sparse, but at least there are some stations with in-situ measurements of BC.

6. The language needs a clean-up. Also the manuscript could improve by focusing on

the main points, and leave out less important content.

Specific Comments:

Page 1, L10: 'Winter warming and sea ice retreat observed in the Arctic in the last decades determine changes of large scale atmospheric circulation pattern (..)' This is controversial. There is a lot of scientific debate whether the Arctic amplification influences the weather at mid-latitudes or not.

Page 2, L29-30: Can you provide references to some of these measurements that you refer to? Also change 'showed' to 'show'

Page3 L6: You need to specify that these are surface concentrations, as Hirdman et al. looked at measurements of EBC at 3 Arctic stations only.

Page 3, L22: 'We estimate the most likelihood BC distribution associated to three large scale atmospheric patterns which mainly contribute the winter near surface warming and sea ice retreat in the Arctic (Dobricic et al., 2016)'. I do not understand this sentence. As I read it you are saying that NAO is the main cause for the Arctic sea-ice retreat?

Section 2.2 needs more references on the atmospheric circulation patterns

Section 2.3: What is the time resolution of the data you are analyzing?

Section 2.3: You refer to many studies comparing your model to observations. Can you summarize the most important points regarding the Arctic and also scavenging?

Page 6, L4: reference to ECHAM5

Page 6, L14: which entire period?

Page 6, L23: Again the entire period. Can you instead specify the period? Is it 1980-2005?

Figure 1: Is this the trend for 1980-2000? Please specify.

Page 7 L12: Did you actually vary the natural variability and the emissions, or is this the reference run 1980-2000? I think this sentence is a bit confusing.

Page 7 L14: How is the wet dep and dry dep calculated?

Page 7 L18: Annually varying anthropogenic emissions are used for the REF simulation, and the BC anthropogenic emissions remained almost constant globally during the simulated period (1980-2005), 4.9 Tg/year, however large changes occurred in those source regions which are also mainly contributing to the transport of BC to the Arctic.' I guess what you mean here is that the global mean emissions of BC have not changed in magnitude during this period. Those source regions, can you specify? I also suggest to move parts of this to Methods, where you discuss the emissions.

Page 7, L25: The positive trend in BC burden is interesting and a bit surprising. You speculate that this is due to increased emissions in East Asia. But why do we then see the negative trend over Eurasia?

Page 7, L 29: 'The natural variability, or the changes in large scale circulation patterns which occurred in the last decades, determined a significant increasing trend of BC dry deposition and surface concentrations over the Arctic, BC wet deposition increased over the Canadian Archipelago and Greenland' Determined? Do you have evidence for this? Again, are you saying that the changes in the circulation patterns determined the change in the wet dep? There are many factors controlling wet dep. The way you are phasing this it sounds like natural variability is the change in circulation patterns?

Page 8, L9: 'Three large scale atmospheric patterns were identified by Dobricic et al. (2016) as the main drivers of the near surface warming in the polar region.' The main drivers for Arctic amplification?

Page 8, L10: Can you say a bit more about these trends? Which months?

Page 8, L121: Do you refer to a particular year for the negative phase of NAO here?

Page 9, L2: 'The BC load has a positive trend over most of the Arctic Ocean, Greenland

and the Canadian Archipelago, and it is related to the presence of the anticyclonic circulation trend over the pole and the cyclonic trend over the North Atlantic extending over Europe.' How do you know this is related, have you showed this? Can you explain this a bit more?

Page 12 L4: 'Different studies found significant connections between the winter sea ice retreat in the Arctic observed in the last decades and changes in the large scale atmospheric circulation.' Which studies do you refer to? Please add this. Also, there are also many studies that have not found any significant connections.

Page 12 L6: What do you mean by 'well approximated', please be more specific. Conclusions: Are you saying here that there is an increase in the blocking frequency over Scandinavia? Again, I miss a discussion here. The number of blocking events and trends detected or not are sensitive to which detection algorithm that is used. E.g. Barnes et al., GRL, 2013 did not find any robust trend in blocking using three different detection methods. The findings you have are based on pure statistical methods (both the ICA ad the MLE), and I think that should be reflected more in the summary and in general.

Technical Corrections:

Page 2, L3: 'has changed' to be consistent.

Page 2, L4: per decade

Page 2,L19: remove 'the' before BC

Page 3, L4: a situation

Page 3, L5: with cloud formation and precipitation with deposition to the surface, do you mean with resulting deposition, because of precipitation?

Page 3, L15: north to North

Page 4 L4 associated 'with' (many places)

Page 6, L4 -and- the tropospheric. . .

---

## Referee Comment (RC2) · Anonymous Referee #3 · 20 Mar 2017

Pozzoli et al. investigated how the black carbon transport and deposition in the Arctic is affected by changing atmospheric circulations. The topic is important given the radiative forcing effects caused by black carbon and other absorptive aerosols could significantly affect Arctic climate. I think this paper should be of great interest to the readers. The simulation experiments and statistical analysis of how the changing atmospheric circulation could affect black carbon transport and deposition in the Arctic is well constructed. Uncertainty and robustness of the statistical analysis is also discussed. However, I think some caveats exist in the link between Arctic warming and changing atmospheric circulation and possible feedback mechanism proposed in this paper. More detailed discussions should be included in order to support this argument.

Interactive
comment

Overall, it is an important study, and should be considered for publication, after the issues mentioned in the reviews have been resolved. Some suggestions for improvements are listed below:

Comments: Page 4, Line 13-15: Please provide more analysis and discussion of how well the Gaussian distribution assumptions hold. Or please cite other references which could support the assumption of Gaussian distribution here.

Page 7, Line 16 - 19: "Annually varying anthropogenic emissions are used for the REF simulation, and the BC anthropogenic emissions remained almost constant globally during the simulated period (1980 - 2005), 4.9 Tg/year, however large changes occurred in those source regions which are also mainly contributing to the transport of BC to the Arctic (Figure S1)." I think this part should include more details, such as a list of names of those regions have large changes in this period?

Page 8, Line 24 - 26: Please specify how the arrows are determined from the trends. Is that a qualitative representation of the tendency of main atmospheric circulation path? Is that path corresponding to black carbon transport path?

Page 9, Line 4 - 7: "This pattern, as the negative phase of the NAO, is characterized by weaker westerlies, colder and drier conditions in Scandinavia and Russia, warmer temperatures over Greenland and Canadian Archipelago with higher precipitations and BC deposition. Dynamically it also imposes stable conditions in which the pollution may accumulate in the polar dome over the Arctic." I think more analyses or references should be provided here to support the argument about changes of atmospheric conditions associated with negative phase of NAO. Also, please clarify the relationship between higher precipitation and BC deposition over Greenland and Canadian Archipelago and the stable condition mentioned later. It's not clear that whether it is most parts of the Arctic becomes more stable except Greenland and Canadian Archipelago? I think additional analysis is necessary for this part.

Page 12, Line 4-5: "Different studies found significant connections between the winter

sea ice retreat in the Arctic observed in the last decades and changes in the large scale atmospheric circulation." Please add reference papers to this section.

---

## Author Comment (AC1) · 12 May 2017

We would like to thank the reviewer for the constructive comments. We have tried to address all of them as detailed below (our answers to each point are formatted in italic, with page and line numbers of the new revised manuscript in squared brackets).

**Anonymous Referee #1**

Pozzoli et al. investigate how changes in three circulation patterns defined in another study influence the transport of black carbon aerosols to the Arctic and the deposition there. The topic is interesting and within the scope of ACP. While the statistical methods the authors use to link the transport of BC with circulation patterns seems valid and well-explained, I am concerned about how the study is presented in general. The authors conclude that changes in atmospheric circulation patterns due to Arctic warming and reduced sea-ice has significantly impacted the BC transport and deposition. For instance, the authors argue that increased blocking over Scandinavia has led to an increase in deposition in West Arctic. The link between Arctic climate change and weather at mid latitudes is controversial, and the authors do not show any attempt to discuss this at all. There is hardly any comparison with previous work or with observations.

Therefore, I cannot recommend publication of this manuscript in its current form.

Below are some recommendations for improvements.

General comments:

1. I think you need to explain the method in Dobricic et al. 2016 in a bit more detail in terms of correlations factors and eigenvalues, as this study is important for your results. How much of the wintertime variability do the patterns explain? How are these patterns linked to sea-ice melting? For instance, you say that: 'The relationship between NAO and SB and the sea ice melting over the Arctic is visible in near-surface temperature trends, the associated warming over the Arctic is the largest over the Barents Sea and west of Greenland.' There is a relationship because you see similar geographical patterns?

*In the ICA methodology, as described in Hyvarinen and Oja (2000), it is not possible to determine the variances of the independent components. We have introduced equation 6 in the new version of the manuscript, where the random matrix **X** (i.e. temporal anomalies in the physical space of an atmospheric variable), can be approximated by the product of two unknown matrices, **A**, with columns containing spatially varying intensities, and **S** with rows containing temporally varying independent components. The magnitude of the independent components in **S** must be fixed, and as they are random variables, the most natural way is to assume that each independent component has variance equal to 1. Then the matrix **A** will be adapted in the ICA solution to take into account this restriction. For the same reason it is not possible to determine the order of the independent components. The main advantage of the ICA method, compared for example to EOF, is that if the underlying climate signals have an independent forcing, we can expect to find loadings (**S**) with interpretable patterns (**A**), whose time coefficients have properties that go beyond simple non-correlation observed by EOFs (Hannachi et al., 2009). We added a more details about the ICA method in the new manuscript to highlight these points [Page 5, L20-30, Page 6, L1-L3].*

*Regarding the second part of the question, we agree with the reviewer that the relationship between sea-ice retreat and mid-latitudes changes was overrated in the manuscript. Surface temperature trends*

*associated to NAO and SB patterns show the largest temperature increase over the Barents Sea and West Greenland, but we cannot conclude on which is the driving process. From Dobricic et al (2016): "By only visualizing patterns it is, however, not possible to evaluate whether the remotely forced anticyclone determines the sea ice melting by advecting the warm air to the northern part of the Barents Sea, as proposed, for example, by Sato et al. (2014), or the sea ice melting defines the near-surface pressure gradient anomalies imposing the formation of the anticyclone, as proposed, for example, by Screen et al. (2013) and Cohen et al. (2014).". As the main focus of this manuscript is the transport of BC associated to mid-latitudes circulation patterns, we have rephrased/removed from the manuscript the discussion about sea-ice and mid-latitude responses.*

2. I think a discussion of the uncertainties using a 'simple' cause-and-effect approach should be included here. The Arctic climate system is complex and noisy, and relationships between e.g. sea-ice retreat and circulation patterns are nonlinear. Many processes may mask each other out. A new perspective paper in Nature Climate Change highlights some of these challenges, and argue that a single cause is unlikely:

http://www.nature.com/nclimate/journal/v6/n11/pdf/nclimate3121.pdf

*We agree with the reviewer's comment, and we have tried to improve our manuscript as suggested. In particular we have modified the text in several parts, including the title, the abstract, introduction and conclusions. We have rephrased throughout the text the statements which implied a simple cause-and-effect relationship between sea-ice retreat and large-scale atmospheric circulation. We have included the suggested reference and other studies which found weak or non-existing correlation between winter sea-ice retreat and mid-latitudes circulation.*

3. In general, I think this manuscript should include more references to previous work.

*We have added and discussed a number of new references (highlighted in red in the new version of the manuscript)*

4. Your conclusions differ from other studies. Hirdman et al. 2010 as you mention in the introduction, finds that changes in emissions were dominant, and that circulation changes only explained a minor fraction of the observed trend. How is your study different? And why do you think they differ so much? Please put your work into context.

*We partly agree with the reviewer on this point. We have shown from a model simulation (REF, Figure 2), that the total trends (with changing anthropogenic emissions and meteorology) of BC surface concentrations and deposition is decreasing over almost the entire Arctic, as shown by Hirdman et al (2010), from observations and back trajectory model results. This is also one of the main points in the conclusions Section. Using model simulations with constant anthropogenic emissions (FIX2000 in Figure 3, and FIX1980 in Figure S2) we have shown how the effect of meteorology may determine trends with different spatial distribution and with similar magnitude. We think that the added value of our study is the estimate with a statistical methodology of the contribution to BC transport to the Arctic trough the specific atmospheric circulation patterns which have shown a statistical significant trend in the last decades. This is discussed in Sections 3.2, 3.3, and summarized in the conclusions. These results are only qualitatively comparable with the work of Hirdman et al (2010), which found significant correlations between the NAO index and EBC variability at two Arctic stations (Alert and Barrow). Their results are nevertheless consistent with ours. The BC transport associated to the $IC_{NAO}$ pattern also show a decrease*

*of surface BC in the Arctic from North Eurasia, and increasing from North America. This is now added in the new version of the manuscript [Page 10, L11-L13].*

5. There are no comparisons with observations at all. The observations in the Arctic are sparse, but at least there are some stations with in-situ measurements of BC.

*We have included in Section 2.3 [Page 7, L16-L35] a long paragraph discussing the performance of the ECHAM5-HAMMOZ model in the Arctic. In particular the results from Bourgeois and Bey (2012), which used the same model version to analyze the transport of pollutants to the Arctic comparing model simulation with ARCTAS campaign. The large bias of simulated BC and EBC concentrations in the Artic is a known issue, shared with several global climate and chemical tranport models (Eckhardt et al., 2015; Sand et al., 2017). In this study we will focus on the impacts of large-scale atmospheric circulation trends on the transport of BC to the Arctic through a novel statistial methodology, assuming that undersestimating BC concentrations does not significantly affect the spatial distribution of the trends.*

6. The language needs a clean-up. Also the manuscript could improve by focusing on the main points, and leave out less important content.

*We have tried to improve the manuscript.*

Specific Comments:

Page 1, L10: 'Winter warming and sea ice retreat observed in the Arctic in the last decades determine changes of large scale atmospheric circulation pattern (..)' This is controversial. There is a lot of scientific debate whether the Arctic amplification influences the weather at mid-latitudes or not.

*We agree with the reviewer on this point and we have rephrased the specific sentence above as well as similar points throughout the manuscript. The new version of the sentence is now [Page 1, L9-L10]:*

*"Winter warming and sea ice retreat observed in the Arctic in the last decades may be related to changes of large scale atmospheric circulation pattern"*

Page 2, L29-30: Can you provide references to some of these measurements that you refer to? Also change 'showed' to 'show'

*We have included references and modified the sentence as follow [Page 2, L30-L34]:*

*"Measurements show that the equivalent BC (EBC, filter based absorption measurements of aerosol particles) surface concentrations in the Arctic, as well as those of other atmospheric pollutants, such as sulphate, are largest in winter and early spring, when the transport of pollutants from lower latitudes is more efficient and the removal processes slower (e.g. Eleftheriadis et al., 2009; Gong et al., 2010; Hirdman et al., 2010; Sharma et al., 2006, 2013). "*

Page3 L6: You need to specify that these are surface concentrations, as Hirdman et al. looked at measurements of EBC at 3 Arctic stations only.

*We have specified that we refer to EBC (see also previous answer) [Page 3, L12-L15]:*

*"Hirdman et al. (2010) analysed the long-term trends of EBC and sulphate measured at three stations in the Arctic. They found that EBC surface concentrations decreased at two Arctic stations, Alert (Canada) and Zeppelin (Svalbard islands, Norway), while there is no trend detectable at Barrow (Alaska)."*

Page 3, L22: 'We estimate the most likelihood BC distribution associated to three large scale atmospheric patterns which mainly contribute the winter near surface warming and sea ice retreat in the Arctic (Dobricic et al., 2016)'. I do not understand this sentence. As I read it you are saying that NAO is the main cause for the Arctic sea-ice retreat?

*We rephrased the sentence as follow [Page 3, L24-L26]:*

*"We estimate the most likelihood BC distribution associated with large scale atmospheric patterns which can approximate the near-surface temperature trend in the Arctic, both spatially and temporally, of two atmospheric reanalysis of the past decades (Dobricic et al., 2016)."*

Section 2.2 needs more references on the atmospheric circulation patterns

*We have largely modified Section 2.2 [Page 5, L20 to Page 6, L22], including a more detailed description of the methodology, the independent component patterns are shown in a new Figure 1, which includes the three patterns with a statistically significant trend of surface temperature (T1000) and geopotential height (H850). To avoid confusion between independent component patterns found in Dobricic et al. (2016) and the well-known atmospheric oscillation indices (NAO, SB, and ENSO), we have renamed the corresponding IC patterns as $IC_{NAO}$, $IC_{SB}$, and $IC_{ENSO}$.*

Section 2.3: What is the time resolution of the data you are analyzing?

*We analyzed monthly mean anomalies, focusing on 3 winter months, December, January and February. This is now specified at the beginning of Section 2.3 [Page 6, L30].*

Section 2.3: You refer to many studies comparing your model to observations. Can you summarize the most important points regarding the Arctic and also scavenging?

*We added the following paragraph [Page 7, L16-L35]:*

*"The model has been extensively evaluated in previous studies by comparing simulated chemical concentrations and physical parameters to observations (Auvray et al., 2007; Pausata et al., 2012, 2013; Pozzoli et al., 2008a, 2008b; Rast et al., 2014; Stier et al., 2005; Bourgeois and Bey, 2012) and within model inter-comparison studies (Kim et al., 2014; Pan et al., 2015; Tsigaridis et al., 2014). As shown by Bourgeois and Bey, 2012, ECHAM5-HAMMOZ largely underestimate BC concentrations over the Arctic, both near the surface as well as in the atmospheric column. Compared to the BC measurements from SP2 instrument [Schwarz et al., 2006; Moteki and Kondo, 2007] of the Arctic Research of the Composition of the Troposphere from Aircraft and Satellites (ARCTAS, Jacob et al., 2010), the simulated BC concentrations show a mean absolute bias in the troposphere of 95% in spring, while in summer the BC is well simulated in the upper troposphere and overestimated by 50% near the surface. Bourgeois and Bey (2012) identified the wet scavenging of aerosol particles as one of the main processes responsible for model bias in winter, a model simulation with revisited wet scavenging coefficients from Henning et al. (2004), considerably improved the simulated BC concentrations in the troposphere in winter, reducing the mean absolute bias to 38%. The large bias of simulated BC and EBC concentrations in the Artic is a known issue, shared with several global climate and chemical tranport models (Eckhardt et al., 2015; Sand et al., 2017). Qi et al. (2017) estimated that the Wegener-Bergeron-Findeisen (WBF) process in mixed-phase clouds ireases BC in the atmosphere by 25% to 70% by reducing wet scavenging efficiency. Other factors which may improve the simulated BC distirbution in the Artcic are dry deposition velocities*

*calculated with resistence-in-series method over all surfaces (ocean, snow/ice) and improved BC flaring emissions (Stohl et al., 2013; Qi et al., 2017). Jiao et al. (2014) shows that BC concentrations in snow are poorly correlated with measurements, and a large spread is found among 25 model simulations, with BC lifetime in the Arctic ranging from about 4 to 23 days, implying large differences in local BC deposition efficiency."*

Page 6, L4: reference to ECHAM5

*Added reference:*

*Roeckner, E., et al. (2003), The atmospheric general circulation model ECHAM5: Part 1, Tech. Rep. 349, Max Planck Inst. for Meteorol., Hamburg, Germany.*

Page 6, L14: which entire period?

Page 6, L23: Again the entire period. Can you instead specify the period? Is it 1980-2005?

*We have specified the period, 1980-2005, where needed.*

Figure 1: Is this the trend for 1980-2000? Please specify.

*Figure 1 of the manuscript is now Figure 2 in the new version. We have clarified the period of the trend.*

Page 7 L12: Did you actually vary the natural variability and the emissions, or is this the reference run 1980-2000? I think this sentence is a bit confusing.

*We refer to the reference simulation (REF) of the period 1980-2005, with annually varying anthropogenic emissions inter-annual meteorological variability. We have rephrased the sentence as follow [Page 8, L26-L28]:*

*"**Error! Reference source not found.** shows the 26-years (1980-2005) trends of BC dry and wet deposition, surface concentration and vertically integrated atmospheric load over to the Arctic for the REF simulation, which includes the effects of both annually varying anthropogenic emissions and inter-annual meteorological variability."*

Page 7 L14: How is the wet dep and dry dep calculated?

*We have included references for the deposition schemes used by ECHAM5-HAMMOZ in Section 2.3 [Page 7, L3-L4]:*

*"The dry deposition scheme of aerosol particles follows Ganzeveld and Lelieveld (1995), while in-cloud and below cloud scavenging follows the scheme described by Stier et al. (2005)."*

Page 7 L18: Annually varying anthropogenic emissions are used for the REF simulation, and the BC anthropogenic emissions remained almost constant globally during the simulated period (1980-2005), 4.9 Tg/year, however large changes occurred in those source regions which are also mainly contributing to the transport of BC to the Arctic.' I guess what you mean here is that the global mean emissions of BC have not changed in magnitude during this period. Those source regions, can you specify? I also suggest to move parts of this to Methods, where you discuss the emissions.

*We have moved this paragraph in Section 2.3 as suggested by the reviewer [Page 7, L7-L12].*

*"The BC anthropogenic emissions remained almost constant globally during the simulated period (1980-2005), 4.9 Tg/year, however large changes occurred in North America, Europe, Former Soviet Union (FSU) and East Asia (Figure S1a,c). Most of the anthropogenic BC is emitted between 30°N and 60°N, and decreased after the 1990s from about 3 Tg/year to about 2.6 Tg/year after 2000. Above 60°N BC anthropogenic emissions are a small fraction of the total and decreased from 100 to 30 Gg/year. The simulation includes also inter-annual varying biomass burning emissions, from tropical savannah burning, deforestation fires, and mid-and high latitude forest fires published by Schultz et al. (2008). BC biomass burning emissions are ranging between 10 and 170 Gg/year above 60°N, and between 35 and 460 Gg/year at mid-latitudes, with peak years connected also to inter-annual meteorological variability (Figure S1b,d)."*

Page 7, L25: The positive trend in BC burden is interesting and a bit surprising. You speculate that this is due to increased emissions in East Asia. But why do we then see the negative trend over Eurasia?

*We don't completely understand the point of the reviewer. We were referring to the trends in the REF simulation. In this simulation the anthropogenic emissions are decreasing in over Eurasia (see also Figure S1 in the supplementary material), so it is not surprising a decreasing trend over this region. While emissions over East Asia are increasing in the same period and BC may be transported to the Arctic by increasing westerlies as described in the new version of the manuscript [Page 10, L17-L21]:*

*"The BC load has a positive trend over most of the Arctic Ocean, Greenland and the Canadian Archipelago, which may be associated with the dipole of pressure anomalies over the Pacific Ocean which is also favouring the export of polluted air masses from East Asia into North America and the Arctic (**Error! Reference source not found.**a). Sharma et al. (2013) previously showed that the contribution of East Asian BC emissions in the Arctic above 200 mb is the largest."*

*See also the answer to the next point.*

Page 7, L 29: 'The natural variability, or the changes in large scale circulation patterns which occurred in the last decades, determined a significant increasing trend of BC dry deposition and surface concentrations over the Arctic, BC wet deposition increased over the Canadian Archipelago and Greenland' Determined? Do you have evidence for this? Again, are you saying that the changes in the circulation patterns determined the change in the wet dep? There are many factors controlling wet dep. The way you are phasing this it sounds like natural variability is the change in circulation patterns?

*We conclude this by comparing the trends of the REF simulation, with varying anthropogenic emissions, and the FIX2000 and FIX1980 simulations, with constant emissions. As this is the only difference between REF and FIX2000 or FIX1980 simulations, we conclude that the trends in model are due to the meteorology. We have rephrased the sentence as follow [Page 9, L11-L15]:*

*"The REF, FIX2000 and FIX1980 simulations are driven by the same atmospheric reanalysis and annually varying biomass burning emissions. The inter-annual meteorological variability of the last decades combined with constant anthropogenic emissions (FIX2000 in **Error! Reference source not found.**, and FIX1980 in Figure S2) determined a significant increasing trend of BC dry deposition and surface concentrations over the Arctic, BC wet deposition increased over the Canadian Archipelago and Greenland."*

Page 8, L9: 'Three large scale atmospheric patterns were identified by Dobricic et al. (2016) as the main drivers of the near surface warming in the polar region.' The main drivers for Arctic amplification?

*We rephrased the sentence as follow [Page 9, L26-L27]:*

*"Three large scale atmospheric patterns were identified by Dobricic et al. (2016) as closely related to the near surface warming trend in the polar region during winter months (DJF)."*

Page 8, L10: Can you say a bit more about these trends? Which months?

*A better description of the trends was included in Section 2.2 [Page 6, L5-L22].*

Page 8, L121: Do you refer to a particular year for the negative phase of NAO here?

*We do not refer to a specific year with negative NAO phase, but at the trend of the $IC_{NAO}$ pattern found by Dobricic et al. (2016) by ICA, which showed a decreasing trend of pressure anomaly over the North Atlantic, and increasing pressure anomaly over the Arctic, similarly to the negative phase of the NAO. We have renamed the patterns found by Dobricic et al. (2016) as $IC_{NAO}$, $IC_{SB}$, and $IC_{ENSO}$, to distinguish them from the known atmospheric oscillations, NAO, SB and ENSO.*

*The general description of the independent patterns was moved from Section 3.2 to Section 2.2, and now reads as follow [Page 6, L9-L22]:*

*"The spatial distribution of the three independent components (ICs) with statistically significant trends were named by visually recognizing their similarity to well-known large-scale weather patterns, the North Atlantic Oscillation (NAO), Scandinavian Blocking (SB), and the El Nino-Southern Oscillation (ENSO). In this manuscript we will refer to the IC patterns as $IC_{NAO}$, $IC_{SB}$, and $IC_{ENSO}$ to avoid confusion with the NAO, SB and ENSO indices. The mean winter H850 trend of $IC_{NAO}$ (**Error! Reference source not found.**a) with increasing geopotential height over the Arctic and decreasing in the Atlantic Ocean near the Azores, clearly appears as a tendency toward the negative phase of the NAO. T1000 increases over the Arctic with maximum over western Greenland, the Canadian archipelago, and the Barents Sea, at the same time it decreases over northern Europe and Siberia (**Error! Reference source not found.**d). The $IC_{SB}$ trend shows an increasing geopotential height over Scandinavia and northwestern Siberia which indicates a prevailing anticyclonic anomaly over the area, bringing warm air from Europe to the Arctic, and vice versa cold air from the Arctic to Eurasia (T1000 in **Error! Reference source not found.**e). As shown in **Error! Reference source not found.**c,f $IC_{ENSO}$ has a small T1000 trend over the Arctic, compared to the other two IC patterns. The reanalysis trends of T1000 and H850 are well approximated by summing the three ICs, all dominant features are captured both spatially and temporally, in particular the prominent dipole between the strong warming in the Arctic and cooling over Siberia (Dobricic et al., 2016)."*

Page 9, L2: 'The BC load has a positive trend over most of the Arctic Ocean, Greenland and the Canadian Archipelago, and it is related to the presence of the anticyclonic circulation trend over the pole and the cyclonic trend over the North Atlantic extending over Europe.' How do you know this is related, have you showed this? Can you explain this a bit more?

*We thank the reviewer for highlighting this inconsistent analysis. We have cleaned the entire paragraph from repetitions and inconsistencies, and now it reads as follow [Page 10, L7-L21]:*

*"The tendency of IC$_{NAO}$ toward the negative phase of the NAO (**Error! Reference source not found.**a) forms an anticyclonic anomaly over the large part of the Arctic Ocean and a cyclonic anomaly in the North Atlantic Ocean. The intensity of westerly winds is decreased in the lower troposphere, with lower transport of pollution from North America across the Atlantic Ocean. On the other hand, the IC$_{NAO}$ slightly increases the transport of pollution from northwest America towards the Arctic Ocean. Consistently with the circulation pathways described in **Error! Reference source not found.**a, the MLEs of BC wet deposition trends related to IC$_{NAO}$ (**Error! Reference source not found.**) show a decreasing trend north of the Eurasian coast and an increasing trend north of America and Greenland. A correlation between the negative phase of the NAO and increasing precipitations and snow accumulation over Western Greenland was also found by previous studies (e.g. Appenzeller et al., 1998; Mosley-Thompson et al., 2005). The BC load has a positive trend over most of the Arctic Ocean, Greenland and the Candian Archipelago, which may be associated with the dipole of pressure anomalies over the Pacific Ocean which is also favouring the export of polluted air masses from East Asia into North America and the Arctic (**Error! Reference source not found.**a). Sharma et al. (2013) previously showed that the contribution of East Asian BC emissions in the Arctic above 200 mb is the largest."*

Page 12 L4: 'Different studies found significant connections between the winter sea ice retreat in the Arctic observed in the last decades and changes in the large scale atmospheric circulation.' Which studies do you refer to? Please add this. Also, there are also many studies that have not found any significant connections.

*We have expanded this discussion at the beginning of the conclusions Section [Page 13, L17-L23]:*

*"The feedbacks between the global warming and arctic amplification with sea-ice retreat and impacts on large-scale atmospheric circulation are still contradictory. The response of mid-latitude weather to the Arctic warming and sea-ice cover changes of the last decades is highly uncertain due to nonlinear processes involved in the Arctic and subarctic climate system (Overland et al., 2016). Some studies find only weak or non-existent relationships between mid-latitude weather structures and Arctic warming (e.g. Screen and Simmonds, 2013; Barnes et al., 2014), while others found correlations between sea-ice retreat in winter over the Barents and Kara Seas and hemispheric scale impacts (e.g. Deser et al., 2007; Petoukhov and Semenov, 2010; Screen et al., 2013; Mori et al., 2014; Di Capua and Coumou, 2016)."*

Page 12 L6: What do you mean by 'well approximated', please be more specific.

*We refer here to Figure 2 and Figure S3 from Dobricic et al. (2016) where it is clear how the trends of reconstructed anomalies depicts all dominant features of the trends in the NCEP and ERA-Interim reanalysis. Also many small scale features were resolved by the reconstructed trends, as well as the temporal evolution, from December to February. See the Figures below from Dorbicic et al., 2016.*

[Figure]

F<small>IG</small>. 2. Colors indicate (top) observed and (bottom) reconstructed trends of T1000 from 1980 to 2015 ($10^{-2}$ K yr$^{-1}$) for (a),(d) December, (b),(e) January, and (c),(f) February. Dots indicate areas with statistically significant trends.

[Figure]

*FIG. S3: Colors: observed (a-c) and reconstructed (d-f) trends of T1000 from 1980 to 2015,  , a), d), December, b), e), January and c), f), February. Dots: areas with statistically significant trends.*

Conclusions: Are you saying here that there is an increase in the blocking frequency over Scandinavia? Again, I miss a discussion here. The number of blocking events and trends detected or not are sensitive to which detection algorithm that is used. E.g. Barnes et al., GRL, 2013 did not find any robust trend in blocking using three different detection methods. The findings you have are based on pure statistical methods (both the ICA ad the MLE), and I think that should be reflected more in the summary and in general.

*We have included more discussion about this point in the manuscript, in the introduction and at the beginning of the conclusions Section.*

Technical Corrections:

*We have corrected all the technical points listed below.*

Page 2, L3: 'has changed' to be consistent.

Page 2, L4: per decade

Page 2,L19: remove 'the' before BC

Page 3, L4: a situation

Page 3, L5: with cloud formation and precipitation with deposition to the surface, do you mean with resulting deposition, because of precipitation?

Page 3, L15: north to North

Page 4 L4 associated 'with' (many places)

---

## Author Comment (AC2) · 12 May 2017

We would like to thank the reviewer for the constructive comments. We have tried to address all of them as detailed below (our answers to each point are formatted in italic, with page and line numbers of the new revised manuscript in squared brackets).

**Anonymous Referee #3**

Pozzoli et al. investigated how the black carbon transport and deposition in the Arctic is affected by changing atmospheric circulations. The topic is important given the radiative forcing effects caused by black carbon and other absorptive aerosols could significantly affect Arctic climate. I think this paper should be of great interest to the readers. The simulation experiments and statistical analysis of how the changing atmospheric circulation could affect black carbon transport and deposition in the Arctic is well constructed. Uncertainty and robustness of the statistical analysis is also discussed. However, I think some caveats exist in the link between Arctic warming and changing atmospheric circulation and possible feedback mechanism proposed in this paper. More detailed discussions should be included in order to support this argument.

Overall, it is an important study, and should be considered for publication, after the issues mentioned in the reviews have been resolved. Some suggestions for improvements are listed below:

Comments:

Page 4, Line 13-15: Please provide more analysis and discussion of how well the Gaussian distribution assumptions hold. Or please cite other references which could support the assumption of Gaussian distribution here.

*We have added the following explanation in Section 2.1 [Page 4, L17-L20]:*

*"In order to simplify the mathematical solution of the problem we assume that the two distributions, p(a|c) and p(c), are approximately Gaussian, meaning that also their product, p(c|a), will be Gaussian. This assumption is supported by the Central Limit Theorem, which tells that the distribution produced by several processes with non-Guassian distributions should appear closer to a Gaussian distribution (e.g. Hyvärinen and Oja 2000). "*

Page 7, Line 16 - 19: "Annually varying anthropogenic emissions are used for the REF simulation, and the BC anthropogenic emissions remained almost constant globally during the simulated period (1980 - 2005), 4.9 Tg/year, however large changes occurred in those source regions which are also mainly contributing to the transport of BC to the Arctic (Figure S1)." I think this part should include more details, such as a list of names of those regions have large changes in this period?

*We agree with the reviewer and we moved the description of the emission scenario in Section 2.3. The new version reads as follow [Page 7, L7-L12]:*

*"The BC anthropogenic emissions remained almost constant globally during the simulated period (1980-2005), 4.9 Tg/year, however large changes occurred in North America, Europe and East Asia (Figure S1a,c). Most of the anthropogenic BC is emitted between 30°N and 60°N, and decreased after the 1990s from about 3 Tg/year to about 2.6 Tg/year after 2000. Above 60°N BC anthropogenic emissions are a small fraction of the total and decreased from 100 to 30 Gg/year. The simulation includes also inter-annual varying biomass burning emissions, from tropical savannah burning, deforestation fires, and mid- and high latitude forest fires published by Schultz et al. (2008). BC biomass burning emissions are*

*ranging between 10 and 170 Gg/year above 60°N, and between 35 and 460 Gg/year at mid-latitudes, with peak years connected also to inter-annual meteorological variability (Figure S1b,d)."*

Page 8, Line 24 - 26: Please specify how the arrows are determined from the trends. Is that a qualitative representation of the tendency of main atmospheric circulation path? Is that path corresponding to black carbon transport path?

*[Page 24] With the arrows in Figure 3 (Figure 1 in the new version of the manuscript) we intended to give a qualitative representation of the tendency of the circulation paths associated to each independent component pattern. Thus, they also represent the tendency of the pollutant transport pathways.*

Page 9, Line 4 - 7: "This pattern, as the negative phase of the NAO, is characterized by weaker westerlies, colder and drier conditions in Scandinavia and Russia, warmer temperatures over Greenland and Canadian Archipelago with higher precipitations and BC deposition. Dynamically it also imposes stable conditions in which the pollution may accumulate in the polar dome over the Arctic." I think more analyses or references should be provided here to support the argument about changes of atmospheric conditions associated with negative phase of NAO. Also, please clarify the relationship between higher precipitation and BC deposition over Greenland and Canadian Archipelago and the stable condition mentioned later. It's not clear that whether it is most parts of the Arctic becomes more stable except Greenland and Canadian Archipelago? I think additional analysis is necessary for this part.

*We thank the reviewer for highlighting this inconsistent analysis. We have cleaned the entire paragraph from repetitions and inconsistencies, and now it reads as follow [Page 10, L7-L21]:*

*"The tendency of $IC_{NAO}$ toward the negative phase of the NAO (**Error! Reference source not found.**a) forms an anticyclonic anomaly over the large part of the Arctic Ocean and a cyclonic anomaly in the North Atlantic Ocean. The intensity of westerly winds is decreased in the lower troposphere, with lower transport of pollution from North America across the Atlantic Ocean. On the other hand, the $IC_{NAO}$ slightly increases the transport of pollution from northwest America towards the Arctic Ocean. Consistently with the circulation pathways described in **Error! Reference source not found.**a, the MLEs of BC wet deposition trends related to $IC_{NAO}$ (**Error! Reference source not found.**) show a decreasing trend north of the Eurasian coast and an increasing trend north of America and Greenland. A correlation between the negative phase of the NAO and increasing precipitations and snow accumulation over Western Greenland was also found by previous studies (e.g. Appenzeller et al., 1998; Mosley-Thompson et al., 2005). The BC load has a positive trend over most of the Arctic Ocean, Greenland and the Candian Archipelago, which may be associated with the dipole of pressure anomalies over the Pacific Ocean which is also favouring the export of polluted air masses from East Asia into North America and the Arctic (**Error! Reference source not found.**a). Sharma et al. (2013) previously showed that the contribution of East Asian BC emissions in the Arctic above 200 mb is the largest."*

Page 12, Line 4-5: "Different studies found significant connections between the winter sea ice retreat in the Arctic observed in the last decades and changes in the large scale atmospheric circulation." Please add reference papers to this section.
*We have expanded the discussion on this point (see also answers to reviewer #1). The first paragraph of the conclusions Section, now reads as follow [Page 13, L17-L23]:*

*"The feedbacks between the global warming and arctic amplification with sea-ice retreat and impacts on large-scale atmospheric circulation are still contradictory. The response of mid-latitude weather to the Arctic warming and sea-ice cover changes of the last decades is highly uncertain due to nonlinear processes involved in the Arctic and subarctic climate system (Overland et al., 2016). Some studies find only weak or non-existent relationships between mid-latitude weather structures and Arctic warming (e.g. Screen and Simmonds, 2013; Barnes et al., 2014), while others found correlations between sea-ice retreat in winter over the Barents and Kara Seas and hemispheric scale impacts (e.g. Deser et al., 2007; Petoukhov and Semenov, 2010; Screen et al., 2013; Mori et al., 2014; Di Capua and Coumou, 2016)."*